# Genome mining yields putative disease-associated ROMK variants with distinct defects

Nga H. Nguyen[1], Srikant Sarangi[2], Erin M. McChesney[1], Shaohu Sheng[3], Jacob D. Durrant[1], Aidan W. Porter[1], Thomas R. Kleyman[3], Zachary W. Pitluk[2]*, Jeffrey L. Brodsky[1]*

1 Department of Biological Sciences, University of Pittsburgh, Pittsburgh, Pennsylvania, United States of America, 2 Paradigm4, Inc., Waltham, Massachusetts, United States of America, 3 Renal-Electrolyte Division, School of Medicine, University of Pittsburgh, Pittsburgh, Pennsylvania, United States of America

☯ These authors contributed equally to this work.
* zachary_pitluk@yahoo.com (ZWP); jbrodsky@pitt.edu (JLB)

**Data Availability Statement:** All relevant data are within the manuscript and its Supporting Information file. Additional data are publicly

## Abstract

Bartter syndrome is a group of rare genetic disorders that compromise kidney function by impairing electrolyte reabsorption. Left untreated, the resulting hyponatremia, hypokalemia, and dehydration can be fatal, and there is currently no cure. Bartter syndrome type II specifically arises from mutations in *KCNJ1*, which encodes the renal outer medullary potassium channel, ROMK. Over 40 Bartter syndrome-associated mutations in *KCNJ1* have been identified, yet their molecular defects are mostly uncharacterized. Nevertheless, a subset of disease-linked mutations compromise ROMK folding in the endoplasmic reticulum (ER), which in turn results in premature degradation via the ER associated degradation (ERAD) pathway. To identify uncharacterized human variants that might similarly lead to premature degradation and thus disease, we mined three genomic databases. First, phenotypic data in the UK Biobank were analyzed using a recently developed computational platform to identify individuals carrying *KCNJ1* variants with clinical features consistent with Bartter syndrome type II. In parallel, we examined genomic data in both the NIH TOPMed and ClinVar databases with the aid of Rhapsody, a verified computational algorithm that predicts mutation pathogenicity and disease severity. Subsequent phenotypic studies using a yeast screen to assess ROMK function—and analyses of ROMK biogenesis in yeast and human cells—identified four previously uncharacterized mutations. Among these, one mutation uncovered from the two parallel approaches (G228E) destabilized ROMK and targeted it for ERAD, resulting in reduced cell surface expression. Another mutation (T300R) was ERAD-resistant, but defects in channel activity were apparent based on two-electrode voltage clamp measurements in *X. laevis* oocytes. Together, our results outline a new computational and experimental pipeline that can be applied to identify disease-associated alleles linked to a range of other potassium channels, and further our understanding of the ROMK structure-function relationship that may aid future therapeutic strategies to advance precision medicine.

available from Github via the URL: https://github.com/mgm68/2023_ROMK_LoF#2023_romk_lof.

**Funding:** This work was supported by grant GM131732 from the National Institutes of Health (NIH) to JLB, by grant DK079307 and DK137329 (Pittsburgh Center for Kidney Research) from the NIH to TRK, by grant DK129285 from the NIH to TRK and S. Sheng, and by award ID 826608 from the American Heart Association to NHN. The funders had no role in study design, data collection and analysis, decision to publish, or preparation of the manuscript.

**Competing interests:** I have read the journal's policy and the authors of this manuscript have the following competing interests: S. Sarangi is an employee of Paradigm4, and Z.W.P. is a former employee of Paradigm4.

## Author summary

Bartter syndrome is a rare genetic disorder characterized by defective renal electrolyte handing, leading to debilitating symptoms and, in some patients, death in infancy. Currently, there is no cure for this disease. Bartter syndrome is divided into five types based on the causative gene. Among these subtypes, Bartter syndrome type II results from genetic variants in the gene encoding the ROMK protein, which is expressed in the kidney and assists in regulating sodium, potassium, and water homeostasis. Prior work established that some disease-associated ROMK mutants misfold and are destroyed soon after their synthesis in the endoplasmic reticulum (ER). Because a growing number of drugs have been identified that correct defective protein folding and/or potentiate ion transport, we wished to identify an expanded cohort of putative disease-associated ROMK mutants. To this end, we developed a pipeline that employs computational analyses of human genome databases along with genetic and biochemical assays. Next, we confirmed the identity of known variants and uncovered previously uncharacterized ROMK variants that are potentially associated with Bartter syndrome type II. Further analyses indicated that select mutants are targeted for ER-associated degradation, while another mutant compromises ROMK function. This work sets-the-stage for continued mining of loss-of-function alleles in ROMK as well as other potassium channels, and may position select Bartter syndrome mutations for correction using emerging pharmaceuticals.

## Introduction

First identified in 1962, Bartter syndrome is group of rare, life-threatening disorders caused by defects in or impaired function of electrolyte channels within the kidney, compromising renal sodium and potassium handling and resulting in excessive electrolyte and water excretion [1]. To date, therapies for Bartter syndrome include electrolyte supplements and non-steroidal anti-inflammatory drugs, which are limited to only mitigating the symptoms. Although disease severity, presentation, and age of onset vary, Bartter syndrome can lead to a failure to thrive, sudden cardiac arrest, and even death [2,3].

One among several causes of Bartter syndrome arises from defects in a potassium channel residing on the apical surface of two segments of the nephron: the thick ascending limb and the cortical collecting duct [4]. The channel, ROMK (also known as Kir1.1), is encoded by *KCNJ1* and was the first inwardly rectifying potassium (Kir) channel identified [5–7]. Like other Kir channels, ROMK functions as a tetramer [8] and exhibits a larger inward current than outward current; all family members also share a common structure that contains two transmembrane domains (TMD) and cytoplasmic N- and C-terminal domains [9]. In the kidney, ROMK plays a central role in mediating potassium efflux, which in turn provides a crucial source of potassium to facilitate sodium reabsorption through the NKCC2 transporter in the thick ascending limb. Furthermore, ROMK-dependent potassium secretion generates a lumen positive transepithelial potential that drives paracellular sodium absorption [10]. Mutations in ROMK give rise to Bartter syndrome type II, also called antenatal Bartter syndrome, since patients often present prenatally (e.g., with excessive amniotic fluid). Among these individuals, observed features include a failure to thrive, renal salt wasting and volume depletion, early post-natal hyperkalemia, hypercalcuria, nephrocalcinosis, and arrhythmias, which together contribute to a high infant mortality rate [11].

In theory, defects in ROMK might arise from a lack of expression, altered protein folding and/or tetramerization, accelerated degradation of poorly folded/assembled subunits, inefficient transport to the cell surface, and/or poor channel (i.e., potassium transport) activity. Indeed, early studies in *X. laevis* oocytes and COS-7 cells demonstrated that some Bartter syndrome type II-associated mutants were absent from the cell surface and others were defective for potassium transport [12–14]. Later work by our group showed that four disease-causing ROMK mutations that cluster in a cytosolic, β sheet-rich immunoglobulin-like domain cause the protein to misfold in the endoplasmic reticulum (ER) [15], an outcome that targets ROMK for ER associated degradation (ERAD).

The ERAD pathway represents a first-line defense in the secretory pathway to recognize and deliver misfolded proteins to the ubiquitin-proteasome system (UPS) in the cytosol. During ERAD, molecular chaperones, such as heat shock protein 70 (Hsp70), recognize and target misfolded proteins for extraction (or "retrotranslocation") from the ER lumen and ER membrane into the cytosol and then for ubiquitination, which serves as a prelude to proteasome-dependent degradation [16–21]. Retrotranslocation requires a $AAA^+$-ATPase, known as Cdc48 in yeast, or p97 (also known as Valosin Containing Protein; VCP) in higher cells [22,23]. In a study utilizing a yeast expression system and human cell lines, we showed that Hsp70 and Cdc48 were required for the degradation of Bartter syndrome-linked mutant ROMK species, whereas wild-type ROMK was relatively stable (15). In addition, the expression of ROMK in a yeast strain lacking two endogenous potassium channels (*trk1Δtrk2Δ*) restored yeast growth on low potassium media [24,25]. As a result, ROMK folding, trafficking to the plasma membrane (where it functions), and potassium transport can be assayed in yeast. Together, these data indicate that the yeast system effectively monitors the efficacy of ROMK biogenesis and provides a facile growth assay, allowing one to screen for defective ROMK mutants in a quantitative and high-throughput manner.

The rapid growth of human genome sequence data and improved curation of existing databases have facilitated the identification of disease-linked genes as well as uncharacterized disease-causing mutations. To date, ROMK mutations associated with Bartter syndrome type II were primarily identified via clinical studies [4,12,26,27], but numerous uncharacterized disease-linked ROMK mutations likely remain unearthed in human databases. We now report on the use of two computational approaches to uncover additional ROMK variants that are potentially associated with Bartter syndrome type II. First, we examined ROMK missense mutations in two NIH-supported databases, the Trans-Omics for Precision Medicine (TOPMed) study [28] and the ClinVar database [29], using an algorithm that predicts mutation severity and pathogenecity. This algorithm, known as Rhapsody [30], utilizes evolutionary conservation along with structural and dynamic features. We previously validated Rhapsody's predictive power to probe the potential impact of both known disease-associated and randomly selected ROMK variants [31]. Second, we performed *in silico* association analyses to identify links between ROMK variants in the UK Biobank and disease-associated phenotypes [32–34] using the REVEAL: Biobank computational platform [35–38]. As a result of these complementary approaches, we report here on the identification and characterization of a cohort of ROMK variants using yeast, *X. laevis* oocytes, and tissue culture cells. Ultimately, we discovered new variants that 1) are unstable and targeted for ERAD, 2) are poorly expressed at the cell surface, and 3) exhibit defective channel function. The identification of a common allele from the two computational approaches validates the complementary nature of these methods and outlines a new pipeline to assess other identified disease-associated mutations in ROMK, an effort that may aid in the development of precision medicines to treat those with Bartter syndrome type II.

## Results

### A computationally-guided analysis reveals uncharacterized ROMK mutations

To isolate previously uncharacterized ROMK mutations associated with Bartter syndrome type II, we first analyzed genomic data collected from the TOPMed program. TOPMed is an NIH-sponsored whole genome sequencing program with a cohort of more than 180,000 participants who have lung, heart, sleep, and blood disorders [28,39]. Moreover, genomic data from the cohort are continuously deposited into the publicly available Bravo browser [39]. At the time of our analysis, a total of 758 ROMK variants were observed in 128,568 individuals, 124 of which are missense mutations (see **S1 Table**). To assess the potential disease severity of each amino acid substitution, we used Rhapsody, a computational algorithm that was first developed to analyze amino acid variants based on sequence conservation, structure, dynamics, and coevolutionary features [30]. The Rhapsody scores, i.e., the predicted pathogenicity, of all 124 missense mutations are also reported in **S1 Table**, and as initially defined, a Rhapsody score > 0.5 suggests a mutation is pathogenic, whereas a benign mutation is assigned a score < 0.5. The accuracy of this method was previously corroborated *in silico* on a dataset of ~20,000 labelled human variants, and when compared with alternative approaches using multiple accuracy metrics, Rhapsody's performance at that time consistently ranked among the highest [30,40]. We further experimentally verified the predictive power of Rhapsody using a yeast growth assay that reports on ROMK plasma membrane residence and channel function (see **Introduction**) [31]. Using the experimental data from the yeast screen, we also previously computed receiver operating characteristic (ROC) curves and found that Rhapsody had the highest accuracy compared to Polyphen-2 [41] and EVmutation [42] (Computed AUROC was 0.86 for Rhapsody, as opposed to 0.81 and 0.77 for Polyphen-2 and EVmutation). Importantly, Rhapsody predicted the severity of known disease-linked ROMK mutations with >90% accuracy [31].

In parallel, we examined potential disease association amongst the 124 TOPMed mutations by cross-examining the NIH ClinVar database, a public archive of human genomic variants and their evidence-based clinical interpretations [29]. Ultimately, we focused on mutations located in regions required for protein folding and function, e.g., the immunoglobulin-like fold and the $PIP_2$-binding domain [43], as well as those designated as having "uncertain clinical significance" for Bartter syndrome in ClinVar [29] (**Fig 1A**).

Based on these analyses, representative mutations were chosen for further assessment (**Fig 1** and **S2 Table**). Most of the mutations (12 out of 17) reside in the ROMK cytoplasmic domain (**Fig 1B**), which contains key regions that play important roles in protein folding and channel function. These regions include the cytoplasmic pore, the G-loop, and the $PIP_2$-binding pocket. For example, T300I is located on a β sheet proximal to the G-loop region, which is solvent-accessible at the top of the cytoplasmic pore and regulates channel gating and inward rectification in ROMK and other Kir channels [44–46]. Given the potential contribution of the T300 site to channel function, we also added T300R, a residue identified in ClinVar, for further analysis.

Ultimately, amongst the 17 variants, 14 were predicted to be deleterious, i.e., assigned a Rhapsody score of ≥ 0.5, with G228E having the highest Rhapsody score (0.930; with the maximum score being 1 [31]; **Fig 1C** and **S2 Table**). Interestingly, G228E resides in the β sheet-rich immunoglobulin domain, in which—as noted above—mutations compromise ROMK folding and stability [15]. It is also noteworthy that the mutation with the highest frequency in the population (0.68%), M357T, is predicted to be neutral. ClinVar predicts that seven mutations are linked to Bartter syndrome, though they are classified as having uncertain clinical

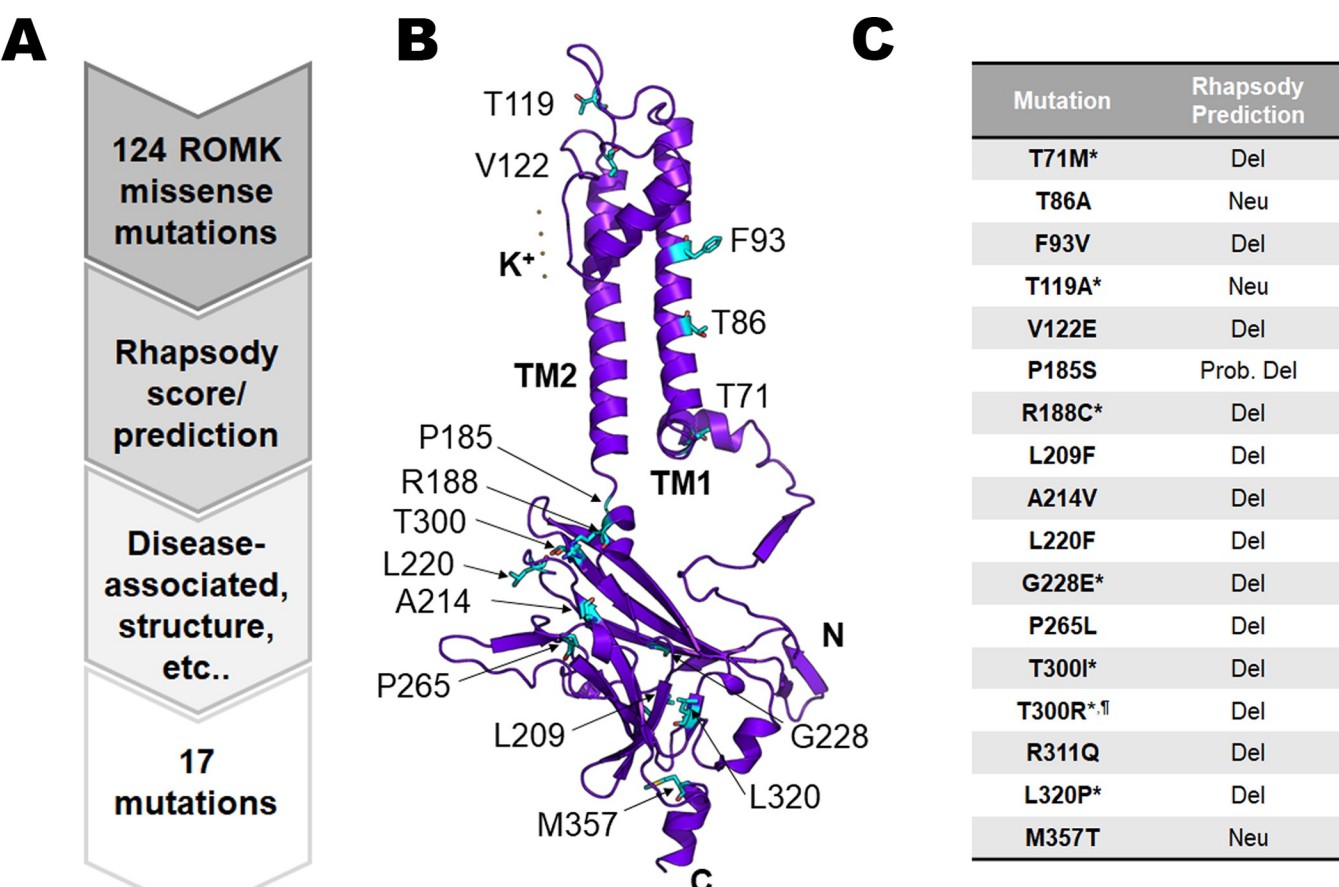

**Fig 1. Computer-guided analysis of TOPMed and ClinVar databases to identify previously uncharacterized ROMK missense mutations that are potentially associated with Bartter syndrome type II.** (A) A flowchart describing how 17 mutations were selected for further examination. All missense ROMK mutations available in the Bravo database [39], in which data from the TOPMed study are continuously deposited, were analyzed. At the time of this study, the Bravo database was in its "freeze 5" version and a total of 124 missense ROMK mutations were available. We then analyzed all 124 mutations using Rhapsody (see text for details) to predict mutation pathogenicity, and picked 16 mutations based on Rhapsody score, disease (Bartter syndrome) association, and location in the ROMK structure. One additional mutation from the ClinVar database (T300R) was also selected. (B) Location of the 17 mutations based on a ROMK homology model. While ROMK tetramerizes to form a functional channel, only one monomer is shown. Seventeen residues of interest are shown as light blue sticks. The homology model was built based on the crystal structure of Kir2.2 (PDB ID: 3SPG), which is 47.42% identical to ROMK1. Images were rendered using PyMOL (ver. 2.6.0). (C) List of 17 mutations and their Rhapsody predictions. "Del" = deleterious, "Neu" = neutral, or predicted to have no effects on channel architecture/function. A designation of "Prob. Del" indicates that the Rhapsody score is close to the 0.5 cutoff. For example, the Rhapsody score of P185S = 0.549 and is thus listed as "Prob. Del". * denotes an uncharacterized Bartter mutation, which is defined as a disease-associated mutation in ClinVar, but is listed as having uncertain clinical significance, ¶ denotes the mutation obtained solely from ClinVar. A comprehensive table of Rhapsody scores and prediction for the 17 mutations of interest, as well as all 124 TOPMed mutations, can be found in S1 and S2 Tables, respectively.

significance (**Fig 1C**, mutations marked with a *). In contrast, seven other mutations were previously associated with Bartter syndrome and have clear clinical consequences (**S2 Table**, denoted by "Bartter" in the "Background information" column). For example, T86A is listed on ClinVar as likely being benign, consistent with its neutral Rhapsody score, while another mutation with a deleterious Rhapsody score, P185S, is in the putative $PIP_2$-interacting domain and disrupts channel conductance, likely by altering $PIP_2$ binding [47]. A comprehensive list of the 17 chosen mutations, along with their Rhapsody predictions, is found in **S2 Table**.

We next assayed the 17 mutants in the yeast growth assay that assesses potassium channel folding, residence at the cell surface, and function. As noted in the **Introduction**, the basis of this assay is that yeast lacking two endogenous potassium channels, Trk1 and Trk2, require the

presence of a functional exogenous potassium channel at the plasma membrane to support growth on low potassium [24,48,49] (**Fig 2A**). We and others previously used the corresponding *trk1Δtrk2Δ* yeast strain to characterize mutant alleles and the channel properties of ROMK and other Kir channels, along with members of distinct potassium channel classes [15,25,50–53].

We expressed the wild-type and each of the 17 mutant ROMK proteins in the *trk1Δtrk2Δ* yeast strain and measured yeast growth in liquid medium in a 96-well plate assay (**Fig 2B**). Yeast containing an empty vector or expressing Y314C, a Bartter mutation previously shown to compromise ROMK folding and function [14,15,54], were used as negative controls, while yeast expressing a related Kir channel (Kir2.1) that traffics more efficiently to the cell surface than ROMK [51,52], along with a known hyperactive ROMK mutant allele (K80M) [55], were used as positive controls. We first noted that G228E, the mutation in the immunoglobulin fold with the most deleterious Rhapsody score, exhibited an expected severe growth defect, i.e., growth was similar to that of yeast containing a vector control (**Fig 2B**). These data are consistent with G228 residing in the β sheet-rich immunoglobulin-like domain (see above and [54]); other disease-associated mutations in this region, including our negative control, Y314C, are rapidly targeted for ERAD, likely due to severe folding defects [15]. In turn, a mutation located at the base of this domain, L320P, similarly prevented yeast growth, though to a lesser extent. These data may reflect the fact that this mutation has a marginal pathogenicity score using Rhapsody (0.588). Based on their growth phenotypes in low potassium, which are reflected by the optical density measured at 600 nm at the end of the yeast growth assay (thus termed "endpoint $OD_{600}$"), we classified the 17 mutants into four groups: severe defect (e.g., G228E), moderate defect (e.g., L320P), slight defect, and no defect. **S3 Table** ("Growth defect categorization" column) summarizes the growth phenotype of each mutation according to this classification, and criteria for these distinctions is also provided in the legend to the Table. In brief, the endpoint $OD_{600}$ values (relative to WT ROMK) used for each category gave rise to distinct growth profiles (**Fig 2**), and the mutant growth levels were easily distinguishable between the break-points in each category. These break-points are as follows: No defect, $OD \geq 1$ (or 100% WT); Slight defect, $0.9 \leq OD < 1$ (90–100% WT); Moderate defect: $0.8 \leq OD < 0.9$ (80–90% WT); Severe defect: $OD < 0.8$ (80% WT).

Importantly, other mutations also exhibited growth phenotypes in accordance with their Rhapsody scores and with previous work. One example is the L220F Bartter mutation, which grew more slowly when expressed in yeast. These data are consistent with a highly deleterious Rhapsody score (0.759), a "pathogenic" classification in ClinVar (**S1 Table**), and previous studies demonstrating reduced channel currents in *X. laevis* oocytes [14,26]. The defect in channel function caused by this mutation likely stems from its localization adjacent to S219, a protein kinase A phosphorylation site [56] that maintains the open state of the channel [57]. In addition, yeast expressing the likely benign and predicted neutral M357T variant grew robustly, as anticipated. Besides M357T, yeast expressing the two other predicted neutral mutations similarly showed minimal or no growth defects (T86A and T119A, see **S3 Table**). Yet, the concordance between Rhapsody predictions and growth phenotypes was not absolute. For example, a Bartter mutation with a "probably deleterious" designation and reported to likely affect $PIP_2$ binding [P185S; [47]] was without consequence in yeast. However, this mutation reduced single channel conductance only when $PIP_2$ was depleted, and had no apparent effect on channel currents or surface expression in *X. laevis* oocytes [47], which might explain the lack of a yeast growth defect.

Although the data outlined above support the predictive power of Rhapsody to report on ROMK residence and activity, we next asked if the dynamic range of the signal-to-noise in these studies might be increased. Therefore, the growth assays with select mutations were repeated using yeast that also expressed the K80M activating mutation (see above). K80 resides

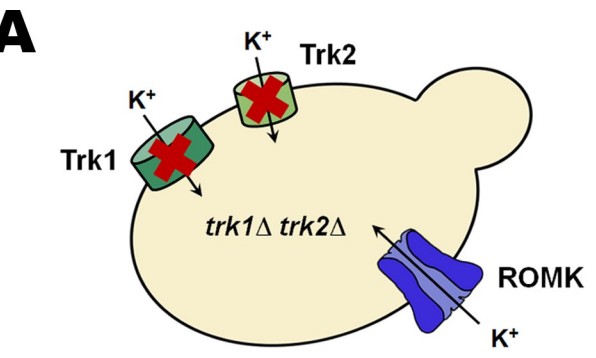

| | High KCl | Low KCl |
|---|---|---|
| **Vector** | Growth | **No growth** |
| **Kir2.1** | Growth | Growth |
| **Wildtype ROMK** | Growth | Weak growth |
| **Neutral ROMK variants** | Growth | Weak growth |
| **Deleterious ROMK variants** | Growth | **No growth** |

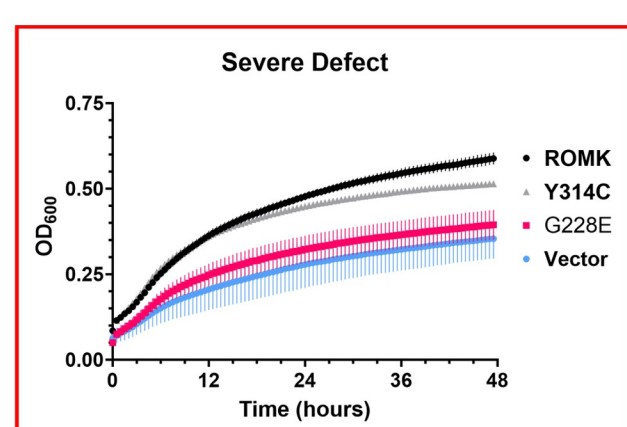

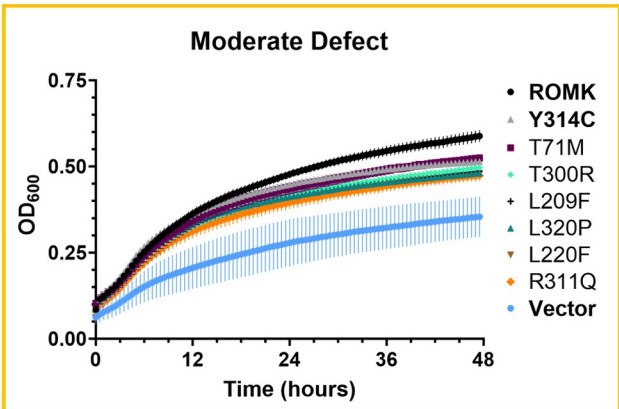

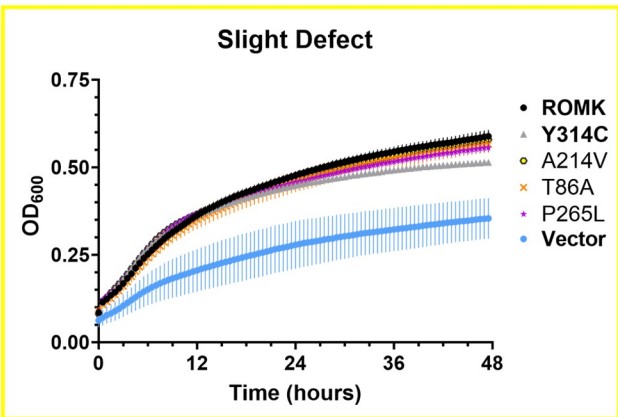

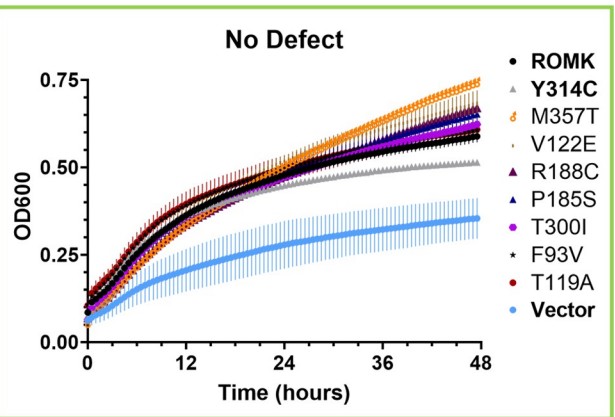

**Fig 2. ROMK mutations from TOPMed and ClinVar show varying growth defects in yeast in low potassium medium.** (A) Schematic of a yeast-based assay to assess the activity of a human potassium channel. A yeast strain lacking endogenous potassium transporters, Trk1 and Trk2, is viable, yet unable to grow on medium containing low potassium, unless a human potassium channel (e.g., Kir2.1 or ROMK) is expressed. Because of impaired ROMK activity at low pH [84] and exaggerated steady-state residence in the ER [51], ROMK exhibits only a weak growth phenotype on low potassium in contrast to Kir2.1. The table shows the expected growth phenotype of yeast containing an empty vector, or expressing a related Kir channel, Kir2.1, or ROMK, and the predicted growth phenotype of yeast expressing a ROMK mutation from **Fig 1**. (B) Viability assays of yeast expressing 17 TOPMed/ClinVar mutations in medium containing low potassium (25 mM) grouped by growth defects. Yeast were transformed with an empty expression vector as a negative control, or with a plasmid expressing Kir2.1, ROMK, or the indicated ROMK mutation. An unstable Bartter mutant (Y314C) [15,54] was used as a negative control. In brief, yeast were grown overnight to saturation and diluted the next day to an $OD_{600}$ of 0.20 with medium supplemented with 25 mM KCl. $OD_{600}$ readings were recorded every 30 min for 48 hrs and normalized to wells containing medium. Graphs were made using GraphPad Prism (ver. 9.5.0), and data represent results from two replicates, ± S.E. (error bars). The growth defect categorization, e.g., "Severe defect", was determined based on the normalized endpoint $OD_{600}$ values at t = 44 hrs and are also described in details in **S3 Table**. Briefly, we determined the break-points for each categorization as follows: No defect, OD ≥ 1 (or 100% WT); Slight defect, 0.9 ≤ OD < 1 (90–100% WT); Moderate defect: 0.8 ≤ OD < 0.9 (80–90% WT); Severe defect: OD < 0.8 (80% WT).

in the putative pH sensor in ROMK and is thought to regulate channel gating [58,59], and previous work in our lab utilized this activating mutation [55] to optimize the signal-to-noise [15,51]. As shown in **S3 Table** (highlighted in red text, and see **S1 Fig**), the dynamic range of the growth assays was improved in yeast expressing select ROMK mutants in the context of the K80M allele. For instance, two mutations (F93V and V122E) predicted to be deleterious by Rhapsody—yet exhibited minor growth defects—now exhibited more measurable growth defects with the improved signal-to-noise in this assay. Furthermore, growth defects for mutations with a "moderate" designation, such as L320P and R311Q, were also now clearly measurable. Nonetheless, most of the remaining mutations exhibited growth defects in accordance with growth assays in the absence of K80M co-expression. These results indicate that more refined data can be obtained when select mutants are examined in the presence and absence of the hyperactive channel, and future efforts will incorporate this paradigm into the experimental plan.

Ultimately, we selected four alleles to characterize at the molecular level. G228E and L320P were chosen for their respective strong (G228E) and more moderate (L320P) growth defects in yeast viability assays and for their status of having clinically uncertain significance in ClinVar. We also selected T86A as a representative neutral mutation that exhibited no growth defect, and T300R due to its moderate growth defect and the importance of the G-loop in supporting ROMK function/stability, as described above.

## Phenotype-guided association analyses of the UK Biobank identified additional disease-associated ROMK variants

In parallel to the data mining protocol above, we pursued an alternate strategy to identify previously ill-characterized and novel disease-linked mutations. More specifically, we wished to identify individuals who exhibit features characteristic of Bartter syndrome type II but are undiagnosed or harbor previously unidentified mutations in *KCNJ1*. Therefore, we utilized the UK Biobank, a genomic and metabolomic resource for multi-omics data retrieved from an ongoing participant study initiated in 2006 [32]. In particular, we performed three genome-wide association studies (GWAS) between phenotypic data and ROMK mutations using REVEAL: Biobank, an analytical platform built upon SciDb [60] that supports elastic scaling for efficient and cost-effective genomic analyses [35–38] (**Figs 3A** and **S2**). We utilized whole exome sequencing data, which at the time of this study, had been made available for ~200,000 UK Biobank participants [61,62]. Whole exome sequencing measures the coding regions of the genome and helps identify disease-causing and/or rare genetic variants. Combined with the large sample size of the UK Biobank cohort and rich phenotypic datasets, whole exome sequencing data can also help elucidate gene function, which is otherwise challenging with imputed genomic data [63–66] and may require the application of additional statistical methods to compensate for missing data [67]. Within the whole exome sequencing dataset, there were 511 *KCNJ1* variants (**S4 Table**), and after applying a minor allele frequency (maf) filter (maf > 1e-05), we selected 142 variants for GWAS.

The first association study employed 25 disease phenotypes for their relevance to ROMK function, to Bartter syndrome type II, and to hypertension [9,68], and included phenotypes such as systolic and diastolic blood pressure, serum urea, creatinine, calcium, and phosphate, as well as urine potassium and sodium. For the second analysis, we selected 15 unique phenotypic codes, or "phecodes", associated with Bartter syndrome type II. The use of phecodes has recently emerged as an effective route to classify clinical phenotypes and is thus suited to phenome-wide association studies compared to traditional billing ICD10 codes [34]. For example, the 15 phecodes we selected represent 25 traditional ICD10 codes. The phecodes chosen

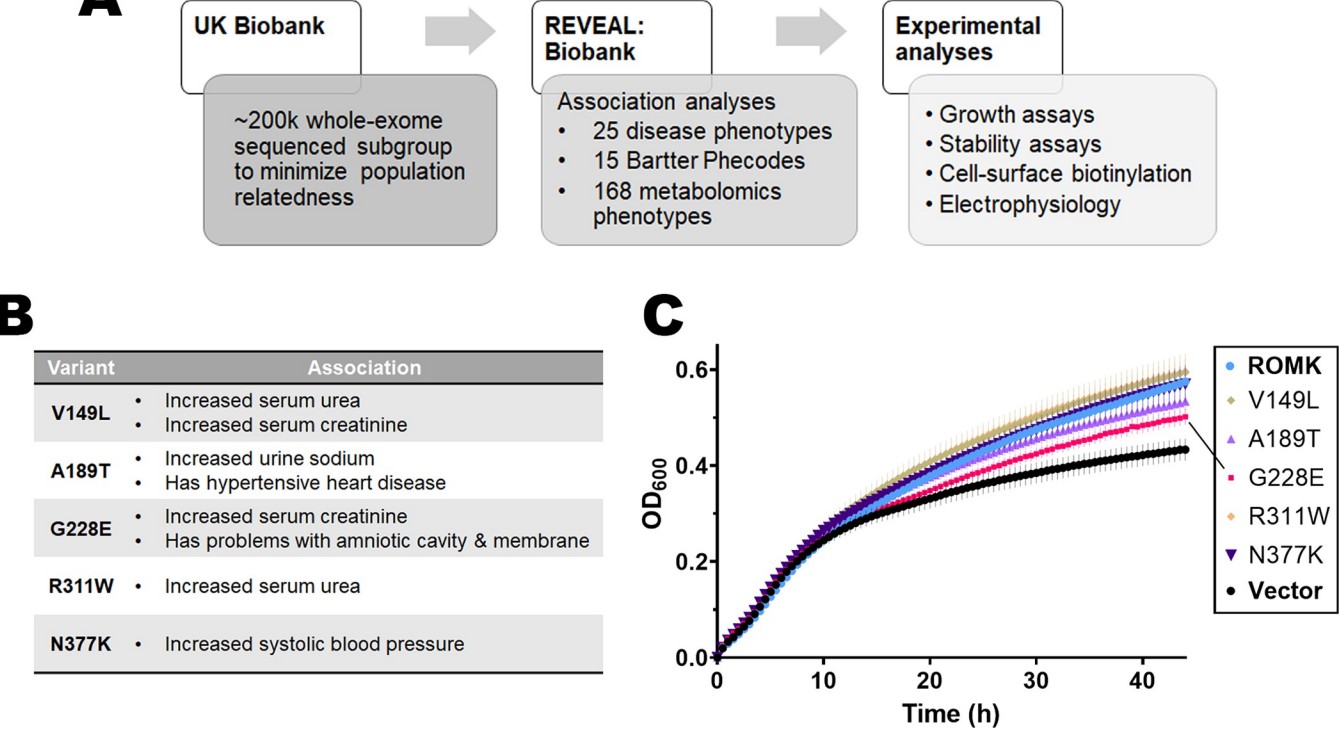

**Fig 3. Mining the UK Biobank to identify ROMK mutations associated with disease-related phenotypes and showing growth defects in yeast.** (A) A flowchart describing how the UK Biobank was mined to search for potential disease-causing mutations. From a sub-population of the UK Biobank (see text for details) that contains genomic and phenotypic data of ~200k participants, we performed *in silico* analysis to find significant associations between ROMK variants and disease-related phenotypes using the computational platform REVEAL: Biobank. (B) Table summarizing the list of potential disease-related ROMK mutations and their associated phenotypes. For a more comprehensive results of the phenotypic association analysis, see **Tables 1–3** and **S6**. (C) Graph shows yeast viability assays in liquid medium supplemented with low potassium (25 mM), as described. $OD_{600}$ readings were recorded over 44 hrs and normalized to the first time point. Graphs were made using GraphPad Prism (ver. 9.5.0), and data represent results from ten replicates, ± S.E. (error bars).

**Table 1. Top significant associations between relevant binary disease phenotypes and ROMK variants in the UK Biobank.**

| SAIGE | | | | | | |
|---|---|---|---|---|---|---|
| **Phenotype** (phecode #) | **Chromosome position** | **Nucleotide change** | **P-value** | **Observation #** | **Mutation** | **Beta value** |
| Problems associated with amniotic cavity and membranes (653.00) | 11: 128839618 | C > T | 6.07E-04 | 122586 | Missense (G228E) | 22.6 |
| Hypertensive Heart Disease (401.21) | 11: 128839736 | C > T | 7.26E-04 | 107442 | Missense (G228E) | 222.96 |
| **REGENIE** | | | | | | |
| **Phenotype** (phecode #) | **Chromosome position** | **Nucleotide change** | **P-value** | **Observation #** | **Mutation** | **Beta value** |
| Hypopotassemia (276.14) | 11: 128839710 | G > A | 7.84E-04 | 121659 | Synonymous (N197N) | 5 |
| Electrolyte imbalance (276.10) | 11: 128839710 | G > A | 9.26E-04 | 121692 | Synonymous (N197N) | 4.8 |

Table shows significant associations obtained from GWAS performed with two algorithms, SAIGE [69] and REGENIE [70]. Relevant disease phenotypes were selected from a list of phecodes, i.e., refined groups of International Classification of Diseases (ICD) codes that are both clinically meaningful and facilitate more efficient genome analysis [34,151]. We defined a binary phenotype as one that is either present or absent in an individual. For example, one either has or does not have "electrolyte imbalance".

**Table 2. Top significant associations between relevant disease phenotypes and ROMK variants in the UK Biobank.**

| SAIGE | | | | | | |
|---|---|---|---|---|---|---|
| **Phenotype** | **Chromosome position** | **Nucleotide change** | **P-value** | **Observation #** | **Mutation** | **Beta value** |
| Urea | 11:128839052 | T > A | 3.86E-05 | 131212 | 3' UTR (-) | 0.02 |
| Phosphate | 11:128839052 | T > A | 5.11E-04 | 120657 | 3' UTR (-) | -0.87 |
| Creatinine | 11:128839856 | C > A | 5.95E-05 | 32083 | Missense (V149L) | 2.48 |
| Urea | 11:128839370 | G > A | 2.93E-04 | 131212 | Missense (R311W) | 0.88 |
| Sodium in urine | 11:128839736 | C > T | 6.10E-03 | 133640 | Missense (A189T) | 1.31 |
| Creatinine (enzymatic) in urine | 11:128839618 | C > T | 9.79E-03 | 133921 | Missense (G228E) | 0.5 |
| Systolic blood pressure (automated) | 11:128839170 | G > T | 6.48E-02 | 130002 | Missense (N377K) | -0.33 |
| Systolic blood pressure (manual) | 11:128839170 | G > T | 9.18E-03 | 7804 | Missense (N377K) | 1.21 |
| REGENIE | | | | | | |
| **Phenotype** | **Chromosome position** | **Nucleotide change** | **P-value** | **Observation #** | **Mutation** | **Beta value** |
| Creatinine | 11: 128839856 | C > A | 9.41E-04 | 131211 | Missense (V149L) | 0.9 |

Table shows significant associations obtained from GWAS with two algorithms: SAIGE [69] for the initial analysis, and REGENIE [70] for result confirmation. 25 relevant disease phenotypes and 168 metabolomic markers were chosen for this analysis (see **Materials and Methods** for details). We termed these phenotypes "quantitative", since they can be measured and compared to wild-type individuals (e.g., serum urea levels).

include clinical Bartter syndrome phenotypes related to fatigue and weakness (e.g., malaise and fatigue), volume loss and thirst (e.g., polyuria), hyperaldosteronism, electrolyte imbalance, and neonatal diagnoses. Notably, the analysis excluded phecodes linked to hypothyroidism and diabetes. We reasoned that these phenotypes might increase the number of false negatives since they are not directly related to ROMK. Finally, we used available metabolomics data from the Biobank for the third association study (**Fig 3A**). A list of all phenotypes examined from the three GWAS is presented in **S5 Table**.

SAIGE [69] and REGENIE [70] were the two algorithms utilized to perform the GWAS. Both algorithms are standards in bioinformatics workflows and identify significant associations, i.e., the computed p-value of a regression test between a mutation and a phenotype in the tested population (see **Materials and Methods**). Results obtained from both algorithms were similar, albeit with minor differences observed in the strength of the association as measured by the p-value(s). The top significant associations identified between *KCNJ1* variants and Bartter-syndrome relevant phenotypes are presented in **Table 1** (for binary phecodes) and **Table 2** (for the quantitative phenotypes and the 168 metabolomic phenotypes). For each association, the name of the phenotype/phecode, the chromosomal position, the base pair change, the computed p-value, the number of participants ("observation #" column), the consequence of the mutation, and the beta value (i.e., effect size) are listed. In addition, the data in **Table 3** represent the mean, median, and standard deviation (S.D.) of the phenotype in individuals carrying a mutation (homozygous or heterozygous) versus wild-type individuals, as obtained from **Table 2**.

**Table 3. Comparison of metabolite levels between wild-type versus individuals carrying ROMK variants.**

| Phenotype | Chromosome position | Units | Mutation | Individuals with mutation | | | | Wildtype individuals | | | |
|---|---|---|---|---|---|---|---|---|---|---|---|
| | | | | # | Mean | Median | S.D. | # | Mean | Median | S.D. |
| Urea | 11:128839052 | mmol/L | 3' UTR | 36744 | 5.45 | 5.32 | 1.38 | 93583 | 5.42 | 5.28 | 1.38 |
| Phosphate | 11:128839052 | mmol/L | 3' UTR | 33846 | 1.16 | 1.16 | 0.16 | 85997 | 1.17 | 1.17 | 0.16 |
| Creatinine | 11:128839856 | mmol/L | V149L | 2 | 0.1 | 0.1 | 0.004 | 32081 | 0.07 | 0.06 | 0.01 |
| Urea | 11:128839856 | mmol/L | V149L | 12 | 6.49 | 6.78 | 1.15 | 131199 | 5.43 | 5.29 | 1.38 |
| Urea | 11:128839370 | mmol/L | R311W | 15 | 6.62 | 6.78 | 1.04 | 131197 | 5.43 | 5.29 | 1.38 |
| Creatinine (enzymatic) in urine | 11:128839618 | umol/L | G228E | 23 | 11863.52 | 12760 | 6323.81 | 133898 | 8724.24 | 7410 | 5680.85 |
| Sodium in urine | 11:128839736 | mmol/L | A189T | 4 | 129.6 | 131.75 | 35.63 | 133636 | 75.22 | 66.4 | 42.96 |
| Systolic blood pressure (manual) | 11:128839170 | mmHg | N377K | 4 | 173 | 168.5 | 23.85 | 7800 | 140.67 | 139 | 19.89 |
| Systolic blood pressure (automated) | 11:128839170 | mmHg | N377K | 27 | 132.74 | 135 | 16.46 | 129973 | 140.07 | 139 | 19.55 |

Disease phenotypes with significant associations with mutations in *KCNJ1* from **Table 2** are listed in the first column. "#" denotes the number of individuals with the mutation or the wild-type allele, and data shown are the means, medians, and standard deviations ("S.D.") of each phenotype.

Interestingly, some of the most significant associations (urea and phosphate, **Table 2**) were linked to a variant in the 3' UTR. We also uncovered a synonymous mutation (N197N) linked to hypopotassemia (hypokalemia) and electrolyte imbalance (**Tables 2–3**). While the phenotypic consequences of these variants was not examined here, future efforts to assay message expression, stability, and/or translation may be meaningful. This is especially important as silent mutations are not always without consequences [71]. For example, a synonymous mutation in the *MDR1* gene alters the folding and function of the resulting protein due to altered codon usage bias [72].

Here, we instead focused on phenotypes with significant associations linked to missense mutations residing in one of the five *KCNJ1* exons, as summarized in **Fig 3B**. Intriguingly, G228E (**Fig 3B** and **Table 1**, and also see above) once again emerged as an uncharacterized, but likely Bartter-associated, mutation, since individuals carrying this mutation exhibited increased serum creatinine and perturbations in amniotic cavity and membrane, both of which are typical manifestations in those with Bartter syndrome type II [73]. This result supports the power of using complementary computational methods and the predictive power of each protocol.

Other mutations from this analysis were also associated with typical Bartter syndrome phenotypes, such as V149L (increased serum urea and creatinine), A189T (increased urine sodium), and R311W (increased serum urea) (**Fig 3B** and **Table 2**; for details on the means of all metabolite phenotypes associated with each mutation, refer to **Table 3**). Another variant at position R311, R311Q, is a known Bartter mutation and was one of the TOPMed mutations that exhibited growth defects (**S3 Table**), which may explain why R311W might similarly lead to impaired ROMK function and disease [14]. Mutations at this residue disrupt inter-subunit salt bridges and pH-dependent gating [59,74]. Finally, we found that heterozygous carriers of N377K had elevated manual systolic blood pressure (**Fig 3B** and **Table 2**), a phenotype atypical of Bartter syndrome since urinary sodium loss usually leads to low blood pressure [75]. This might be attributed to the fact that blood pressure is a complex and multi-genotypic trait [76], or possibly due to errors in the manual measurement of blood pressure in these individuals, which is not uncommon [77,78]. However, hypertension has been observed in a clinical case study in which a newborn presenting with classical antenatal Bartter syndrome phenotypes (i.e., renal salt wasting and hyperkalemia) also had transient high blood pressure [79]. Subsequent genetic testing revealed that the infant carried two mutations in the *KCNJ1* gene, E151K

and a deletion of amino acids 116–119, which again strongly supports a diagnosis of antenatal Bartter syndrome. Moreover, the N377K mutation was also identified in the TOPMed program (**S1 Table**) and was therefore chosen for further analysis. Finally, it is worth noting that all individuals with these five mutations (V149L, A189T, G228E, R311W, and N377K) are heterozygous carriers (**S6 Table**), which could contribute to minimal or undiagnosed disease. In addition, the mutations are rare, with each occurring fewer than 27 times in the 200,000-person cohort (**S6 Table**).

To assay for any potential functional defects caused by these five mutant alleles, we measured the growth of yeast expressing each variant in liquid medium (**Fig 3C**). G228E again showed a growth defect in *trk1Δtrk2Δ* yeast on low potassium, consistent with its highly deleterious Rhapsody score. Another predicted-deleterious mutation at the base of the transmembrane domains, A189T (Rhapsody score 0.644), similarly slowed yeast growth, albeit to a somewhat lesser extent. Meanwhile, V149L, a mutation in the extracellular domain that organizes the potassium selectivity filter, grew as well as the wild-type control, perhaps reflecting its low Rhapsody score (0.193). In contrast, robust growth of yeast expressing the remaining two mutations (R311W and N377K) was observed in low potassium-containing media. The Rhapsody scores for these two mutations are 0.781 and 0.287, respectively. (Note that the structure at N377 is absent from the homology model, so an independent Rhapsody analysis of this mutation was performed using a predicted ROMK monomeric model obtained from Alpha-Fold [80]; also see **Discussion**). In contrast, the lack of a growth defect in yeast expressing R311W was surprising since previous studies in *X. laevis* oocytes showed that this mutation reduced channel currents [14,81]. Perhaps the discrepancy can be attributed to the difference in intracellular pH in the two systems, which is pH 4–5 in yeast [82] and pH ~7.5 in *X. laevis* oocytes [83]; ROMK is known to be pH sensitive, exhibiting maximal channel opening at pH 7.8, and the majority of the channels are closed upon a shift to pH ~6.6 [84,85]. However, the actual role of R311 in channel function remains disputed. pH gating was initially thought to be mediated by the formation of an RXR triad (R41, K80, R311) [81], but subsequent structural studies cast doubt on the formation of the triad, and instead favored a model in which R311 formed intermolecular salt bridges with E302 from an adjacent monomer [74]. Finally, even though N377K appeared to exhibit wild-type-like growth, the yeast ODs vary greatly across the replicates (**Fig 3C**, error bar of dark purple line), which is consistent with stochastic toxic effects of this mutation (see **Discussion**). Ultimately, given their strong associations with Bartter syndrome phenotypes, we selected all five mutations to characterize further.

## A subset of the newly uncovered, putative disease-associated ROMK mutations destabilize the protein

Prior work established that potassium channel variants can lead to disease by interfering with channel conductance, open probability, or abundance at the cell surface [86]. The last of these possibilities is regulated by cellular protein quality control pathways, which monitor the folding state of a protein both in the ER—which may lead to ERAD—or in later steps of the secretory pathway, which may lead to lysosome targeting [87–89]. Since some of the identified mutations reside in the cytoplasmic domain (namely, G228E, T300R, R311W, L320P, and N377K), which includes the critical immunoglobulin-like region [see above and [15,54]], we surmised that these mutations would decrease protein stability. To test this hypothesis, we measured protein stability via a cycloheximide chase analysis in *trk1Δtrk2Δ* yeast expressing each variant [90].

As shown in **Fig 4A**, the G228E protein was highly unstable compared to wild-type ROMK, with almost no protein remaining after 60 minutes. These results were expected given: 1) the

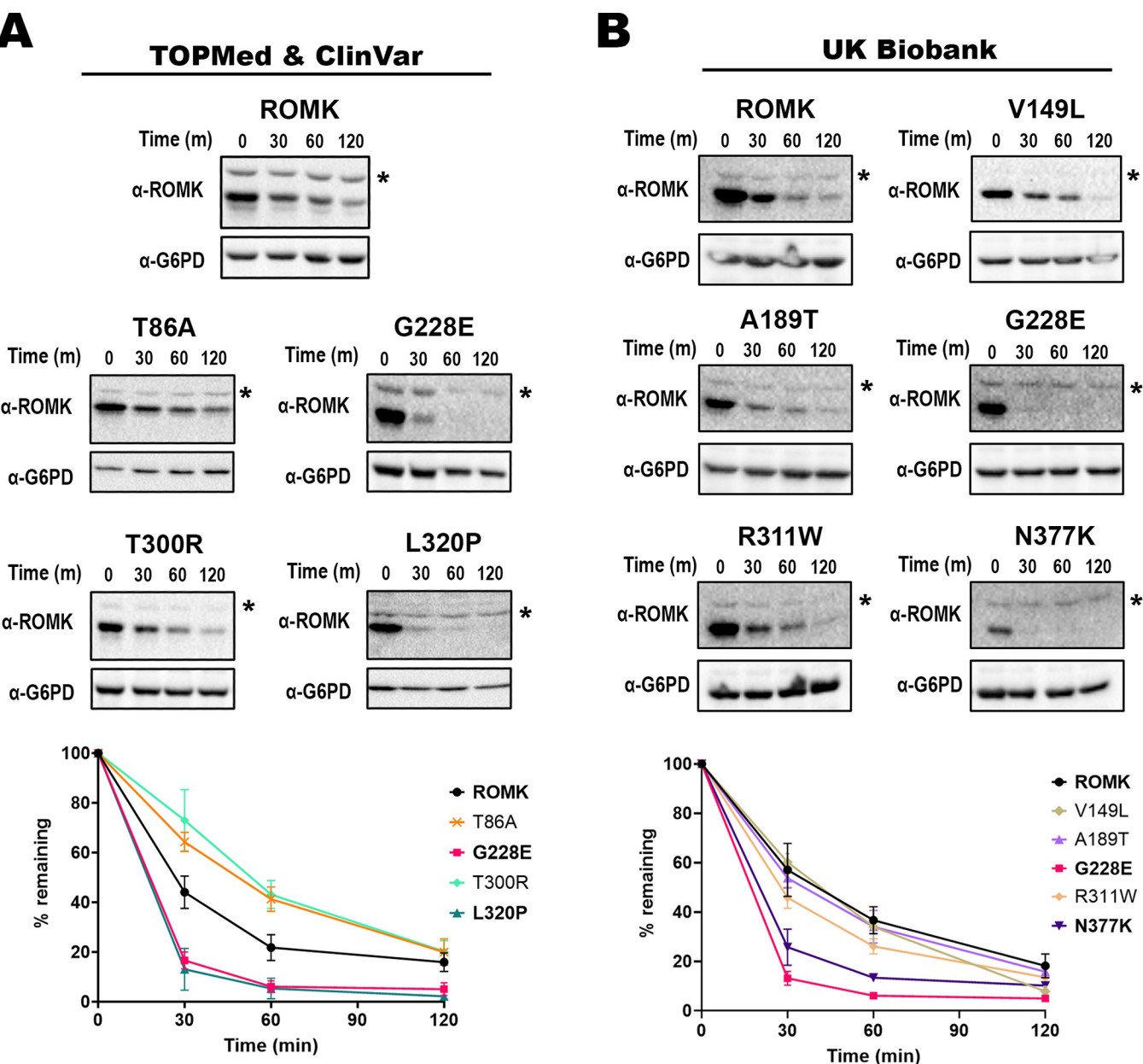

**Fig 4. Select disease-linked mutations in the gene encoding ROMK destabilize the protein.** Stability assays were performed in *trk1Δtrk2Δ* yeast expressing different ROMK variants retrieved from (A) TOPMed/ ClinVar and (B) the UK Biobank databases. In brief, yeast cultures were grown to mid-log phase, diluted, and incubated at 37˚C for 30 min before cycloheximide was added. Cultures were then collected at the indicated time points and processed for immunoblot analysis. A rabbit antiserum was used to detect ROMK [160], and a rabbit monoclonal antibody against G6PD was used as a loading control (see **Materials and Methods**). Representative immunoblots are shown, and graphs show the percentage of the protein remaining over time, compared to the 0 min (m) time point, as quantified using ImageJ (ver. 1.53c). * indicates a non-specific protein band recognized by the ROMK antiserum, so only the bottom bands in the ROMK immunoblots were used for the quantification. Graphs were made using GraphPad Prism (ver. 9.5.0), and data represent the means of at least three independent experiments, ± S.E. (error bars). For each experiment, a representative immunoblot is shown.

severe growth defect observed in these cells after incubation in low potassium (**Fig 2B** and **S3 Table**), 2) the change from glycine to a bulky charged amino acid (glutamic acid), which given the residue's location in the β-sheet rich region might have drastic consequences on protein structure, and 3) the highly deleterious Rhapsody score (0.930). Another mutation from

TOPMed, L320P, which as indicated above resides in a β sheet at the base of this domain, also significantly destabilized the protein, almost to the same extent as G228E (**Fig 4A**). These results are consistent with its severe-to-moderate growth defects in yeast, the conversion of large hydrophobic residue to a proline, and a Rhapsody score that predicts a deleterious outcome (**Fig 2B** and **S3 Table**). It is also intriguing that the analogous residue (L321) in Kir2.1 resides in an amino acid patch (SYLANEI**L**W) that binds AP-1 and promotes Golgi export [91,92], but this Golgi export consensus sequence is absent in ROMK, suggesting that the structural change instead triggers premature degradation. In contrast, neither T86A nor T300R destabilized ROMK. In fact, the T86A and T300R substitutions appeared to modestly stabilize ROMK (**Fig 4A**). For T86A, these results are consistent with its assignment as being neutral in Rhapsody (score: 0.063), and with robust growth observed in the yeast assay. On the other hand, it was somewhat surprising that the T300R protein was stable, despite a Rhapsody score of 0.722 and the moderate growth defect in yeast (**Fig 2B** and **S3 Table**). We hypothesized that the mutation might instead alter a key architectural feature associated with ROMK function rather than overall stability (see below).

Among the UK Biobank mutations, only N377K—besides G228E—appeared to compromise protein folding (**Fig 4B**), despite its lack of effect on yeast growth. Even though N377K did not result in a marked defect in yeast growth, there was a significant difference in growth phenotypes across replicates (**Fig 3C**), suggesting the acquisition of spontaneous suppressors [93] (also see **Discussion**). In any event, the net growth phenotype of N377K is consistent with its designation of a neutral mutation by Rhapsody (0.287).

Together, 12 out of the total 16 predicted deleterious mutations exhibited some degree of growth defects in yeast, especially when growth assays in the presence and absence of the K80M allele are taken into consideration (see above). Similarly, all five predicted neutral mutations (N377K included) were without or with only minimal defects. Second, the accuracy of Rhapsody is enhanced for mutations with high Rhapsody scores, as evidenced by the fact that 9 out of 10 mutations with Rhapsody scores >0.7 impaired yeast growth. On the other hand, the correlation between Rhapsody scores and protein stability is weaker, consistent with Rhapsody simply predicting overall pathogenicity. Third, our results indicate that data mining efforts identify previously uncharacterized ROMK mutations that destabilize the protein, an outcome that may contribute to disease presentation and positions these mutations—and certainly other newly uncovered mutations—as targets of therapies that may one day restore ROMK folding, as seen for other protein conformational diseases [94,95].

## Select ROMK mutants are targeted for ERAD and limit ROMK levels at the cell surface

Some ROMK variants that significantly alter structure (e.g., Y314C; **Fig 2**) are targeted for ERAD, as shown previously [15]. Therefore, we next asked if the ERAD pathway is also responsible for the accelerated degradation rates observed in **Fig 4** for the G228E, L320P, and N377K alleles. To this end, we again performed stability assays in yeast, as described above, but in this case protein turnover was measured in the presence or absence of MG-132, a drug that inhibits the chymotrypsin-like activity of the proteasome [96]. To augment the effects of MG-132, these experiments were performed in the *pdr5Δ* yeast strain that lacks a multidrug efflux pump [97], as extensively employed in previous studies [see e.g., [15,53,98]]. Consistent with ERAD targeting, protein stability assays revealed that all three mutant proteins were subjected to proteasome-dependent degradation (**Fig 5**, compare DMSO and MG-132 results). As shown previously [15], the wild-type channel was also targeted for proteasome-dependent degradation, but to a lesser extent (compare relative stabilities of wild-type versus the mutants in

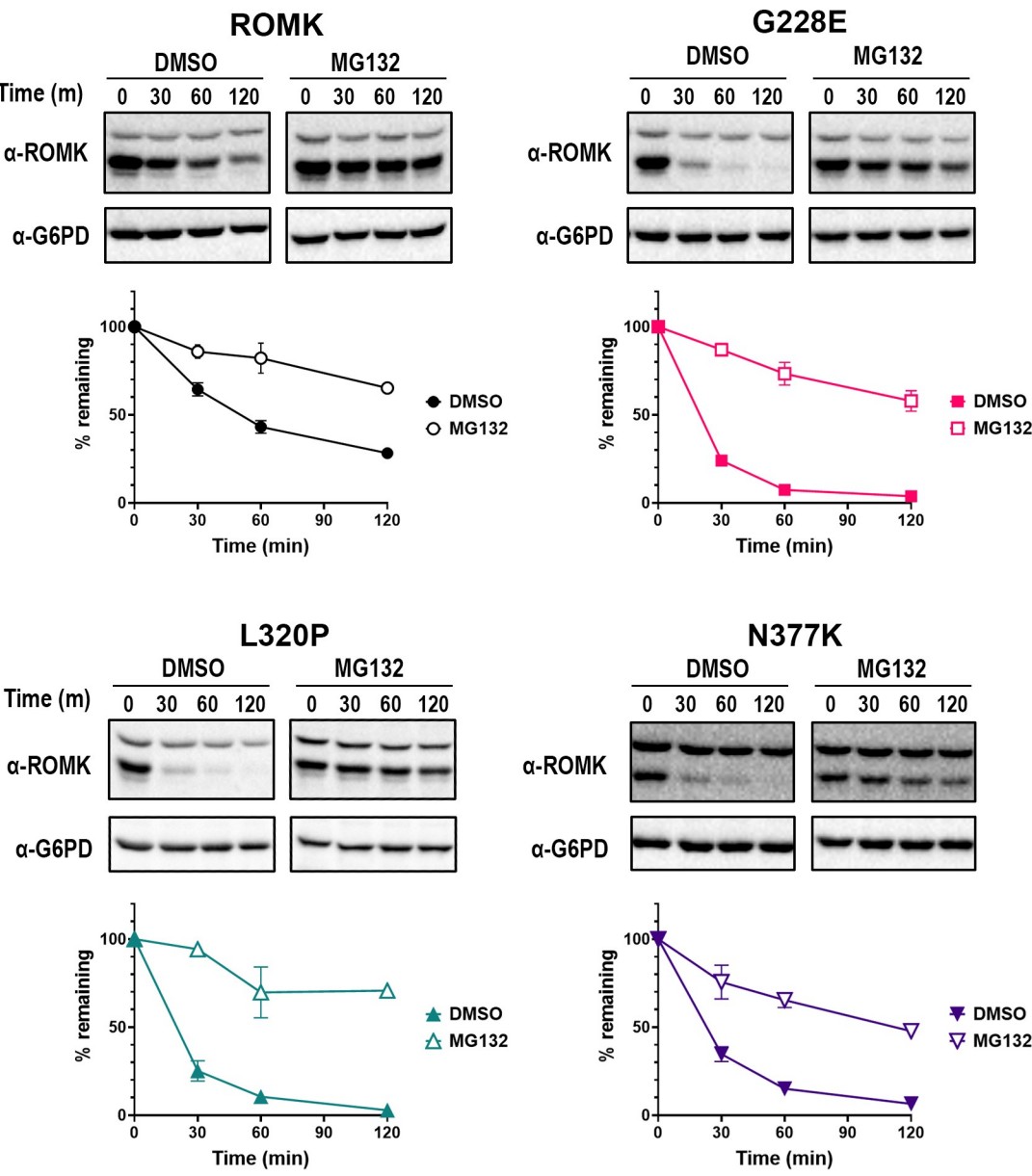

**Fig 5. Three disease-associated ROMK mutants are degraded by the proteasome in yeast.** Stability assays were performed in *pdr5Δ* yeast expressing wild-type ROMK, or ROMK carrying the G228E, L320P, or N377K mutation. Yeast cultures were grown to mid-log phase, diluted, and incubated at 37°C for 30 min with either the proteasome inhibitor MG-132 or an equal volume of the vehicle (DMSO) before cycloheximide was added. Cells were collected at the indicated times, processed, and immunoblot analysis was performed as described in the **Materials and Methods**. Representative immunoblots are shown, and graphs show the percentage of the protein remaining over time, compared to the 0 min (m) time point, as quantified by ImageJ (ver. 1.53c). Similar to **Fig 4**, only the bottom bands in the ROMK immunoblots were used for the quantification. Graphs were made using GraphPad Prism (ver. 9.5.0), and data represent the means of at least three independent experiments, ± S.E. (error bars). For each experiment, a representative immunoblot is shown.

the DMSO control). This is likely due to the inefficient and inherently error-prone process of protein folding and channel assembly in the ER, and has been seen frequently for other ion channels, such as ENaC [99], CFTR [100], hERG [101], and Kir2.1 [53].

To confirm that the three disease-associated mutants that underwent proteasome-dependent degradation are selected for ERAD, we next conducted stability assays in a temperature

sensitive yeast strain, *cdc48-2* [102], which encodes a defective temperature-sensitive allele of *CDC48*, the gene encoding the $AAA^+$-ATPase that mediates protein retrotranslocation in yeast [103,104]. At a non-permissive temperature, each of the ROMK mutant proteins was again significantly stabilized (**S3 Fig**), providing further evidence that these mutations send the protein for premature degradation via the ERAD pathway.

To confirm these data in a more physiologically relevant cell system, we conducted stability assays in HEK293 cells transfected with each ROMK variant and again used MG-132 to inhibit the proteasome. G228E, L320P, and N377K were degraded to variable extents when compared to wild-type ROMK (**Fig 6A**). Specifically, G228E and N377K exhibited somewhat higher

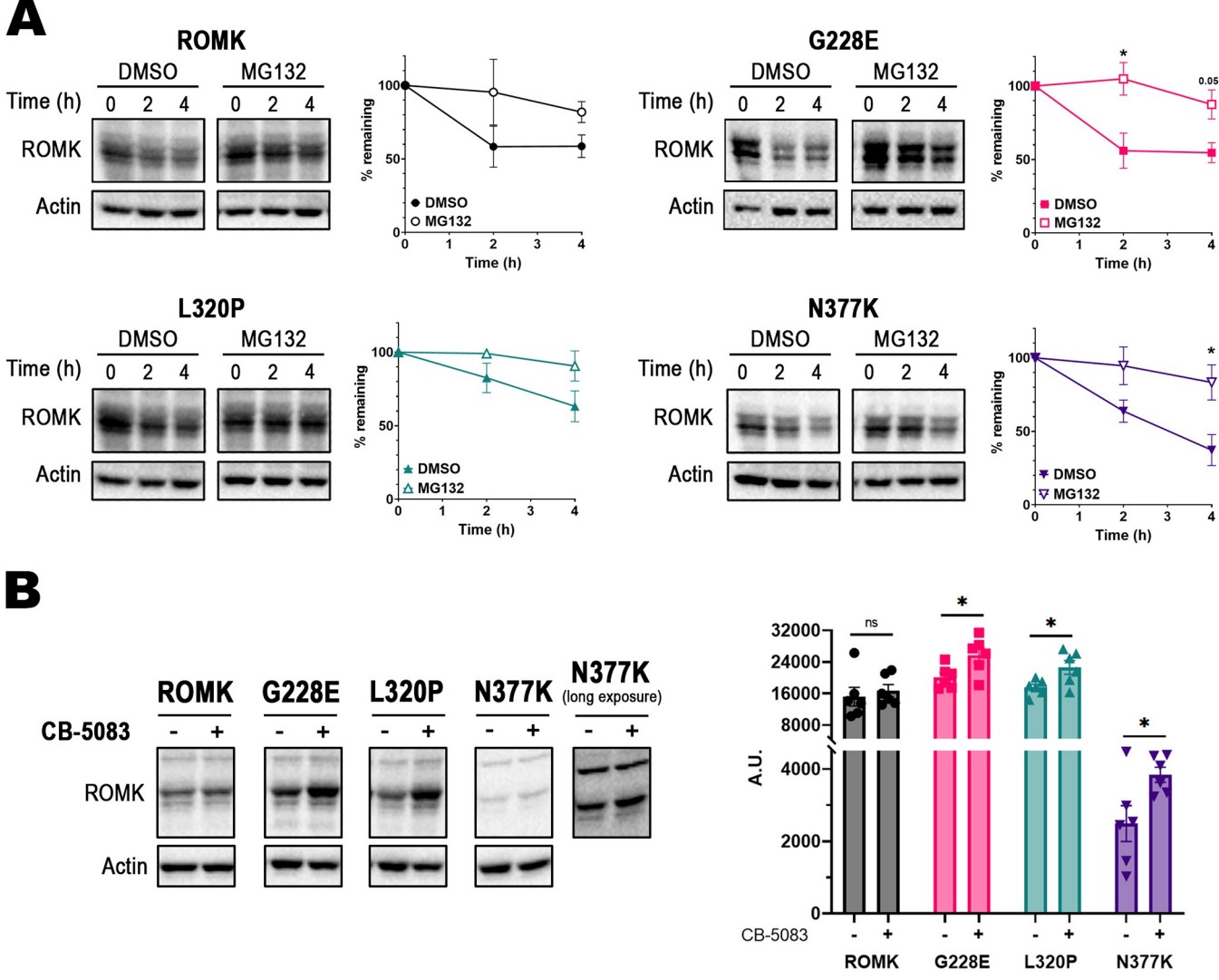

**Fig 6. The ERAD pathway also contributes to ROMK mutant turnover in HEK293 cells.** (A) Stability assays of HEK293 cells expressing wild-type ROMK, or ROMK carrying the G228E, L320P, or N377K mutation. HEK293 cells transfected with the indicated expression vector were treated with MG-132 or the equivalent volume of DMSO for 30 min, at which point cycloheximide was added. Cells were next processed as described in the **Materials and Methods**. Representative immunoblots are shown, and graphs show the percentage of the protein remaining over time, compared to the 0 hr (h) time point. A rabbit antiserum was used to detect ROMK [160], and a mouse monoclonal antibody against actin was used as a loading control. All bands present in the ROMK immunoblots are specific for the protein and were used for the quantification. (B) Steady-state protein levels before and after p97 inhibition with CB-5083 for 4 hrs. Graphs were made using GraphPad Prism (ver. 9.5.0), and data represent the means of at least three independent experiments, ± S.E. (error bars). For each experiment, a representative immunoblot is shown, and the quantification was performed using ImageJ (ver. 1.53c). Since the top-most band in the ROMK immunoblots represents a non-specific protein species recognized by the ROMK antiserum, the bands in the center of the blots were used for the quantification. p-values in were calculated with two-tailed Student's t-test for independent samples. ns, $p \geq 0.05$; *, $p < 0.05$.

degradation rates, with 55% and 37% of the protein remaining, respectively, by the end of the 4-hr experiment (compared to 59% for the wild-type protein). In contrast, L320P was only mildly unstable (again compare the relative curves in the presence of the DMSO control). In fact, the degradation rate of L320P (63% remaining) was nearly identical, if not slightly improved, compared to that observed for wild-type ROMK (see **Discussion**). However, the degradation of each protein was slowed in the presence of MG-132, again consistent with ERAD targeting.

We subsequently assessed ERAD in the presence or absence of an inhibitor of p97, which is the mammalian homolog of Cdc48 [105–107]. The compound, CB-5083 [108], is somewhat toxic and has been used in clinical trials for various cancers [109]. Thus, we performed steady-state measurements of ROMK after treatment with CB-5083 or the DMSO control, as employed previously [15]. As shown in **Fig 6B**, a statistically significant increase in the G228E, L320P, and N377K mutant proteins was evident, whereas the levels of the wild-type protein in the presence or absence of CB-5083 were less dramatically affected.

Although the N377K protein was unstable (**Fig 6A**), we routinely observed significantly less protein at steady-state and in the degradation assays at the 0 min time point (compare matched and long exposures of N377K and the other mutants, as well as the wild-type protein, in **Fig 6B**). Based on these results, we surmise that the N377K mutation either triggers abortive translation, or the protein product is rapidly degraded co-translationally, leaving a sub-pool that then turns over more slowly by ERAD. Each of these scenarios has been observed as a source of the molecular etiology underlying other diseases [110–112]. While a full definition of this phenomenon awaits further analysis, this outcome might in principle result in Bartter syndrome type II. We also cannot rule out the possibility that this minimal pool of N377K channels at the plasma membrane might still provide sufficient potassium transport, which could account for the lack of a growth defect in the yeast assays in **Fig 3**. By comparison, the level of measurable Kir2.1 channels at the yeast surface required to support the growth of *trk1Δtrk2Δ* cells in low potassium represents <20% of the total protein pool [52].

It is important to highlight that the overall muted level of protein destabilization observed in HEK293 versus yeast cells is consistent with the fact that the ERAD pathway in yeast is hyperactive. Similar results with misfolded mutant alleles in ROMK and Kir2.1 have been observed previously [15,53]. It is also worth noting that the mutation with the most modest Rhapsody score (0.588), L320P, exhibited the most wild-type-like degradation phenotype in HEK293 cells.

The enhanced dependence on p97 to maintain the steady-state levels of the G228E, L320P, and N377K mutants in HEK293 cells suggests that lower levels of these proteins should reside at the cell surface. To test this hypothesis, we expressed the wild-type protein and the ROMK variants in HEK293 cells and performed cell-surface biotinylation assays to measure the plasma membrane protein pool [51,113]. As anticipated, markedly lower levels of biotinylated G228E and L320P channels were observed at the plasma membrane relative to the wild-type protein (**Fig 7**). In addition, and as noted above, the levels of N377K were significantly lower in HEK293 cells, so the biotinylated protein pool at the cell surface was also drastically reduced (**S4 Fig**, note lanes 4 and 8 in the immunoblot). As controls for labeling specificity, the Na$^+$/K$^+$-ATPase—a plasma membrane resident—was identified after avidin pull-down of the biotinylated material, whereas Hsp90, an abundant cytosolic protein, was absent. Taken together, these data indicate that disease-associated mutations identified from the complementary genomic databases deplete ROMK at the cell surface, which likely contributes to disease.

## T300R abolishes channel activity

In contrast to changes in protein stability, protein deficiency, and/or altered abundance at the cell surface, disease-associated mutations in ion channels might traffic normally but are unable

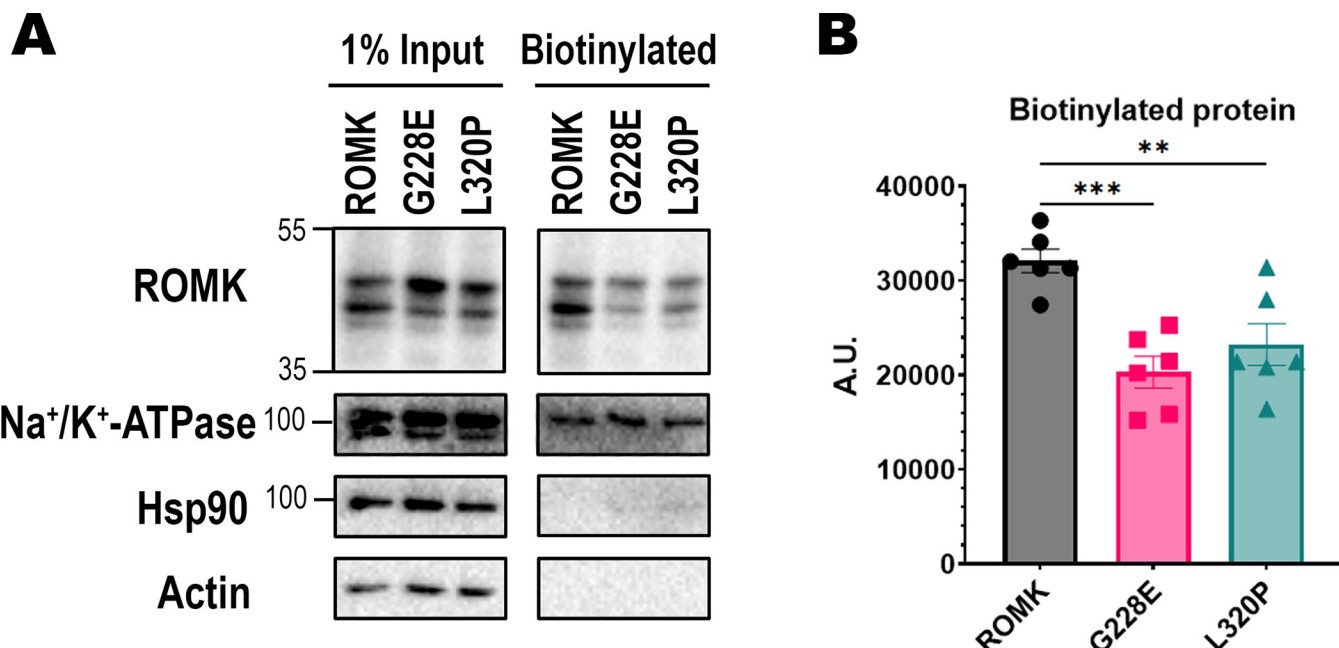

**Fig 7. Cell surface levels of putative disease-associated ROMK mutants are reduced in HEK293 cells.** (A) A cell-surface biotinylation assay is shown to indicate the relative surface expression levels of the indicated ROMK variants. HEK293 cells expressing wild-type ROMK or the G228E or L320P mutant were treated with biotin, processed, and incubated with streptavidin beads before an immunoblot analysis was performed. 1% input was collected prior to the overnight incubation, while the "Biotinylated" material represents precipitated cell surface protein. A representative immunoblot is shown, with a rabbit antiserum to detect ROMK [160], a mouse monoclonal antibody against the $Na^+/K^+$-ATPase, a mouse monoclonal antibody for Hsp90, and a mouse monoclonal antibody against actin. (B) Graph shows the quantification for biotinylated protein, as measured by ImageJ (ver. 1.53c). All bands in the ROMK immunoblots were used for the quantification, and the N377K variant was omitted, as its protein levels were significantly lower. Error bars represent the means of six independent experiments, ± S.E. p-values were calculated with two-tailed Student's t-test for independent samples. ns, $p \geq 0.05$; *, $p < 0.05$; **, $p < 0.01$; ***, $p < 0.001$.

to support ion conductance, as observed for class III mutations in CFTR [114]. Characterizing this phenotype is vital as—in contrast to the ERAD-targeted F508del CFTR protein repaired by chemical chaperones—the class III mutant defects can be treated with approved potentiators [115,116]. For ROMK, the mechanisms of gating are under active investigations [9,44], but the general consensus is that channel gating capitalizes on the helix-bundle crossing region, $PIP_2$ binding, and a narrow opening at the top of the cytoplasmic pore, known as the G-loop, as described above [9,43]. Therefore, we focused on one of the mutations identified from the ClinVar database, T300R, which is located on the G-loop. As shown above, this mutant compromised the growth of the *trk1Δtrk2Δ* yeast strain in low potassium (**Fig 2B** and **S3 Table**), suggesting defective potassium transport, yet the protein was stable (**Fig 4**). Similar observations were made in a previous study in which two ROMK mutations, P185S and R188C, moderately increased protein surface expression, yet negatively affected channel gating and conductance in a $PIP_2$-dependent manner [47]. In addition, a homologous mutation in the closely related Kir2.1 channel, M301R, prevented channel function [45]. Therefore, we predicted that T300R would also reduce channel currents.

To measure channel activity, two-electrode voltage clamp assays were performed in *X. laevis* oocytes expressing wild-type and select ROMK variants. As hypothesized, the T300R mutation completely abolished ROMK current (**Fig 8A and 8B**), reducing ROMK-specific, i.e., barium sensitive, currents to the same level as the negative controls (i.e., oocytes injected with water or expressing the Y314C mutant [see above and [14,15,54]]). Consistent with these data,

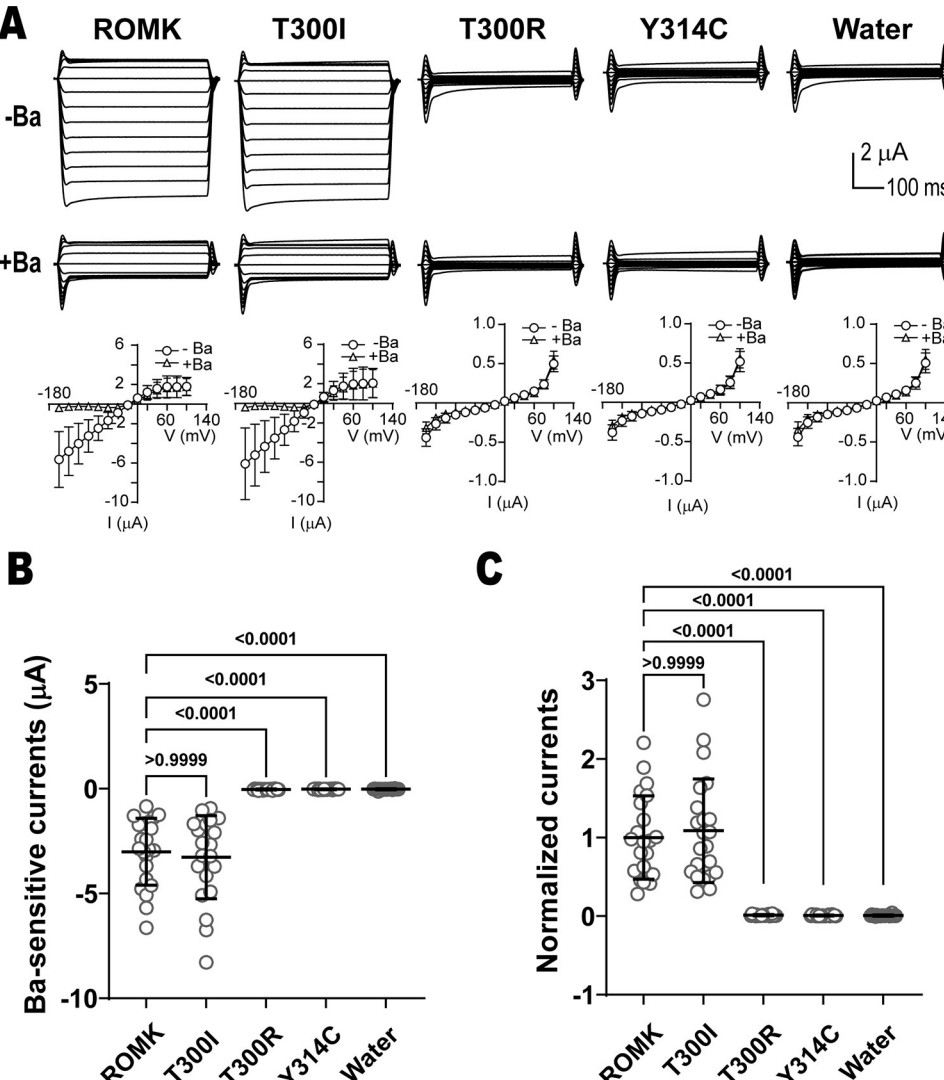

**Fig 8. The T300R mutation in ROMK abolishes channel currents.** (A) Top panel: Currents recorded by two-electrode voltage clamps (TEVC) in *X. laevis* oocytes. Oocytes from female *Xenopus laevis* were injected with 1 ng of the indicated cRNAs, or the equivalent volume of water. 20–30 hr following cRNA injection, TEVC recordings were measured at different voltages (-160 mV to 100 mV, in 20 mV increments) in a bath solution containing 50 mM KCl (for more details, see **Materials and Methods**). Currents were recorded in the presence or absence of 1 mM $BaCl_2$ and I-V plots are shown (bottom). In addition to a water-injected control, the results with a known unstable disease-causing mutant (Y314C) are shown [15,54]. (B) Graph shows the $Ba^{2+}$-sensitive ROMK current in oocytes injected with the indicated conditions, as recorded by TEVC. (C) Normalized currents, which are defined as $Ba^{2+}$-sensitive currents divided by the means of the wild-type currents. Error bars in the graphs in (B) and (C) show the means of 22 replicates, ±S.D. p-values (shown above the data) were computed using Kruskal-Wallis and Dunn's multiple comparisons tests. The data shown are a representative result from three independent experiments using three batches of oocytes.

structural modeling of T300R suggests that the change from a small hydroxyl into a large basic side chain likely occludes the cytoplasmic pore and prevents potassium passage (**S5 Fig**).

Because another mutation at the same site, T300I, was one of the 17 alleles identified from the TOPMed and ClinVar databases (**Fig 1**), we also examined currents corresponding to this variant. In contrast to the effect of the T300R allele, the current was identical to wild-type ROMK when oocytes were injected with a cRNA for T300I ROMK. The wild-type-like current

is perhaps expected given the lack of a growth defect in T300I-expressing yeast (**Fig 2B** and **S3 Table**) as well as the less consequential substitution from one beta-branched amino acid to another. Thus, in contrast to the unstable mutants that are absent at the cell surface (e.g., G228E and L320P), these results shed light on a functional defect associated with a stable putative Bartter syndrome-associated ROMK mutant. More generally, these data highlight the power of uniting a computational analysis and the yeast system as an initial read-out to screen ill-characterized and previously undefined alleles in a potassium channel-encoding gene.

## Discussion

In the kidney, efficient plasma filtration and electrolyte reabsorption are achieved through a system of transporters and ion channels [117], among which ROMK plays a crucial role. Potassium efflux through ROMK in the thick ascending limb and the cortical collecting duct of the kidney nephron helps maintain potassium and sodium homeostasis [9,118]. Over 40 missense mutations in the gene encoding ROMK, *KCNJ1*, have been identified and linked to Bartter syndrome type II [4], a rare autosomal recessive disease presenting with fluid loss and electrolyte imbalance, i.e., renal salt wasting, polyuria, early post-natal hyperkalemia and subsequently hypokalemia [119]. Previous investigations of the cellular mechanisms of Bartter-associated ROMK mutations have primarily focused on variants that affect whole-cell currents [12,120], and each study commonly analyzed a handful of mutations. Thus, for both this disease and most other protein conformational diseases, there exists a need to systematically identify potential disease-causing variants in the genome, especially with the increasing availability of human genomic and phenotypic data from large-scale worldwide studies [121]. To this end, recent efforts dedicated to the systematic assessment of missense variants have incorporated massively parallel sequencing (VAMP-seq) [122,123] and deep mutational scanning [124].

In this study, we utilized two computationally-guided approaches to mine three human genomic databases (TOPMed, ClinVar, and the UK Biobank) with the ultimate goal of identifying novel and previously uncharacterized mutations that are potentially associated with Bartter syndrome type II. From the initial analyses, 21 mutations were selected for expression and functional screening in the established *trk1Δtrk2Δ* yeast system [24,25], among which one mutation (G228E) was identified from both approaches. Based on results from yeast viability assays, we again validated the ability of the Rhapsody algorithm to develop predictions of mutation severity. Specifically, we found that 17 out of 21 mutations exhibited growth phenotypes in accordance with their Rhapsody scores, i.e., 12/16 that were scored as deleterious exhibited strong growth defects in the yeast system and 5/5 scoring as neutral were largely without effect. In addition, as highlighted in the **Results**, Rhapsody was more effective for scores >0.7, with an accuracy of 90%, compared to 81% when all predicted deleterious mutations were considered. To further assess how well Rhapsody can distinguish deleterious from non-deleterious mutations, we calculated a receiver operating characteristic (ROC) curve based on results from yeast growth assays for the TOPMed/ ClinVar mutations. Consistent with previously published ROC data [31], we found that the area under this curve (AUROC) is 0.7625 (**S6 Fig**), again indicating that Rhapsody can effectively identify deleterious mutations.

It bears mention that while Rhapsody demonstrates a high accuracy in predicting mutation pathogenicity [30,40], this method relies on the availability of a protein structure/homology model, which has become more attainable thanks to recent advancements in AI-assisted protein structure prediction exemplified by AlphaFold [80]. Regardless, the reliability of a predicted structure must be evaluated before using the structure for calculations with Rhapsody. Yet, for low-confidence structures, the analysis can be complemented by comparing pathogenicity scores from other methods, such as Polyphen-2 [41] or EVE [125]. Moreover, our

analysis of human variants using Rhapsody yielded mutant alleles that destabilize the ROMK protein, but it is important to reemphasize that Rhapsody was not specifically designed to predict changes to protein stability. The development of computational methods to assess protein stability has significantly progressed, and notably a recent method was optimized for membrane proteins [126]. However, there is still a considerable level of inaccuracy and/or limited accessibility with these methods [127]. Given that protein destabilization is a prevalent cause of inherited diseases [128], the need for an accurate yet accessible and comprehensive computational method to detect destabilizing mutations is paramount. In any case, we believe that a well-rounded *in silico* assessment of mutation pathogenicity, coupled with follow-up functional assays, remains a powerful approach to identify and characterize new variants. To this end, we further validated the accuracy of Rhapsody predictions by cross-referencing other computational tools (namely, Polyphen-2 [41], EVmutation [42], and EVE [125]). These results are summarized in **S7 Table**.

Because most prior functional analyses of ROMK variants focused on those that impair channel function [12,120], we specifically sought mutations that compromise protein folding and trafficking. To this end, we conducted functional assays in yeast, *X. laevis* oocytes and HEK293 cells, thereby revealing distinct cellular mechanisms underlying potential disease etiology. One newly identified and previously uncharacterized Bartter mutation (T300R) had no effect on protein stability but blocked channel conductance. In contrast, three mutants (G228E, L320P, and N377K) were unstable in yeast (which exhibit a hyperactive ERAD pathway), with more varying degrees of stability in mammalian cells, as reported for studies on other ERAD substrates [15,53].

G228E, which was uncovered from both screening strategies, likely affected protein folding due to the substitution of a small, aliphatic amino acid with a larger charged residue. This effect is also consistent with the mutation's localization in the β sheet-rich cytoplasmic domain. Despite decreased differences in protein degradation rates between the wild-type protein and the G228E mutant in mammalian cells, reduced cell surface expression was observed. In line with these findings, whole-cell currents in *X. laevis* oocytes expressing this mutant were indistinguishable from those in the water injected control (**S8 Fig**). Perhaps unsurprisingly, these results are consistent with our finding that two individuals heterozygous for G228E from the UK Biobank exhibited issues with their amniotic cavity and membrane, a typical manifestation of antenatal Bartter syndrome [75,119].

Another Bartter mutation of uncertain clinical significance, L320P, also destabilized the protein in yeast, yet there was little effect on stability in HEK293 cells. Since L320P is also located in the immunoglobulin domain, where thus far seven ROMK mutations compromise protein stability [15,54], we reasoned that the folding of this domain in the ER is a rate-limiting step, at least in yeast. Despite the lack of an effect on protein stability in mammalian cells, we still observed significantly reduced protein at the cell surface. This fact, coupled with its wild-type-like protein level at steady-state, suggest that the L320P mutation affects ROMK trafficking at later steps in the secretory pathway, a process that may then be rate-limiting in higher cells [129].

While the third mutation, N377K, initially appeared to lack a growth defect in yeast, there were significant stochastic effects between experiments (note the larger error of these measurements in **Fig 3** compared to the other strains). When whole-cell currents of oocytes expressing this mutant were measured, the absence of a significant defect was upheld, but there appeared to be a small reduction in the measured current (**S8 Fig**, compare the means of the currents between N377K and wild-type). Nonetheless, the N377K mutant protein was rapidly degraded in yeast through the ERAD pathway, and to a lesser extent in mammalian cells. Curiously, Rhapsody designated a neutral score for this mutation (0.287), perhaps reflecting the limitation

of this program in analyzing mutations that rely on structural predictions (i.e., AlphaFold) instead of homology models. (Please note that the use of AlphaFold was imperative since N377 lies beyond the sequence that was resolved in structural studies). It is also possible that the amino acid substitution alters a critical post-translational modification or allostery, which Rhapsody is unable to capture. This possibility is supported by the discovery of a nearby residue, N375, that resides within a non-canonical endocytic signal (YxNPxFV) that binds to the ARH adaptor and recruits ROMK to clathrin-coated pits [130]. This model is also consistent with our proposal, above, that later steps in the trafficking pathway are altered.

Although with some faults, the *trk1Δtrk2Δ* yeast model provides a rapid, inexpensive, and quantitative route to screen mutations that affect potassium channel folding, trafficking to the cell surface, and function. Because this system is also amenable for drug discovery [25], future work will attempt to rescue variants whose defects were confirmed in higher cells (e.g., G228E). Yet, discrepancies between defects in yeast growth, protein stability, and/or confounding results in higher cells—as seen for N377K—hint at variables that must be taken into account in future screens. Based on the growth of transformants on plates that displayed a range of colony sizes, as well as the larger errors seen in growth assays, the N377K mutation may cause toxic effects on yeast growth, which results in the accumulation of spontaneous suppressors [93] and the formation of "petite" colonies (see for example [131,132]). Indeed, spontaneous suppressors arising from mutations in hexose or amino acid transporters are observed as common causes for phenotypic reversion in *trk1Δtrk2Δ* yeast [133–135]. To test the latter possibility, we propagated cells from colonies of yeast expressing the N377K mutant on medium containing a nonfermentable carbon source (i.e., glycerol) instead of glucose. We found that the smaller colonies failed to grow on plates containing glycerol (S7 Fig), a phenotype typical of the so-called "petite" yeast [136] that arises due to spontaneous mutations in, or the loss of, its mitochondrial DNA [137,138].

To mine the UK Biobank data, we utilized REVEAL: Biobank, a high-performance, cost-effective computational platform to explore, query, and analyze multi-omic biobank-scale datasets [35–38]. REVEAL: Biobank's ability to rapidly filter a large search space to create cohorts of interest, execute complicated bioinformatics workflows at scale in a user-friendly manner, and allow custom algorithms (e.g., phecode generation) for ease of application positions REVEAL: Biobank as an optimal solution for high-throughput *ad hoc* analysis. Moreover, multiple algorithms, such as SAIGE [69] and REGENIE [70], can be incorporated into the workflow with simple parameter changes. This allows results to be validated, which is vital given discrepancies frequently observed in bioinformatics tools.

In addition to the correlative associations obtained from GWAS, the beta value, i.e., effect size, provides a powerful measure of the degree and direction of impact that a mutation has on the phenotype. While more work is needed to verify the degree of the impact observed, the direction of the beta values (+/-) in Table 1 follows the direction of the difference seen in the mean values of the phenotypes between the wild-type and mutant genotype cohorts in Table 3.

Results obtained from the UK Biobank GWAS analyses largely corroborated the findings from the yeast screen and Rhapsody predictions, but it is interesting to note that the p-value of the associations were lower than those deemed significant in typical GWAS (1e-08). A probable explanation for the low p-values could be the high imbalance observed in the ratios of cases (i.e., individuals with a phenotype) and controls, and of wild-type and mutant genotypes (see Tables 3 and S6). This highlights the need to employ multiple approaches of hypothesis testing and validation as well as the potential limitations of *in silico* models. Thus, future efforts might also utilize metrics other than the p-value to determine significance [139]. Finally, it is worth noting that individuals harboring these mutations are almost certainly heterozygous carriers.

Consequently, deciphering phenotypic presentation and obtaining meaningful p-values from GWAS are even more challenging.

In preliminary studies, we expanded our efforts to investigate the heterozygosity of disease-associated mutations in experiments using TEVC in *X. laevis* oocytes (**S8 Fig**). Interestingly, we observed an intermediate effect in oocytes expressing both the wild-type and the mutant alleles, i.e., the currents were approximately half of wild-type (51% for WT/G228E and 46% for WT/T300R). Not only do these data support the notion that heterozygosity leads to phenotypic differences, but they also provide a possible explanation for why we found significant associations between disease phenotypes and individuals carrying the G228E allele even though they are likely heterozygotes.

The pipeline using REVEAL: Biobank described in this paper can also be expanded into two directions to further dissect the cellular and biochemical mechanism underlying ROMK function and to elucidate the relationship between ROMK and other diseases. In the first direction, we can use other known phenotypes associated with ROMK, such as hypertension, to uncover additional mutations that exert a functional effect on ROMK trafficking or function. This approach can also be extended to include linkage disequilibrium calculations coupled with burden tests to identify co-occurring mutations in proteins known or thought to interact with ROMK, essentially identifying synthetic interactions, but not necessarily synthetic lethal interactions [140,141]. We previously obtained these outcomes with the cytoskeletal scaffold protein encoded by *SLC9A3R2* [142]. The entire set of mutations could then be fed into an artificial intelligence (AI)-based application, such as the AlphaFold Protein Structure Database [80], to provide insights into the structural implications of the mutations. The second direction is a bootstrapping approach to uncover potential new disease connections by leveraging both genetic and health record data to explore longitudinal prescription and general practitioner information (i.e., READ codes) for patients with identified mutations in key genes [143–145]. Consequently, there is ample opportunity to further explore ROMK/*KCNJ1*, and other putative disease-linked genes, by leveraging large human datasets in the UK Biobank. While definitive links between the variants we identified and disease presentation await further study, computational predictions of disease linkage will undoubtedly improve as these datasets expand.

In sum, our work highlights a pipeline for computational-guided mining of human databases to search for mutations in any potassium channel that can be assayed in yeast. We identified and then uncovered the cellular mechanisms underlying potential disease phenotypes in a subset of ROMK mutations with uncertain clinical significances, among which three destabilize the protein and one is channel-defective. This is especially important for the development of therapeutic strategies, i.e., the use pharmacological chaperones for misfolded mutants versus potentiators for channel-defective alleles [146,147]. It is worth noting, however, that numerous uncharacterized uncharacterized ROMK mutants remain, and new disease-associated variants will continue to arise. Future work should thus focus on improving the output and signal-to-noise of the yeast assay so that more mutations can be simultaneously screened, which combined with studies in higher cells may ultimately contribute to the development of precision medicine to treat those with Bartter syndrome type II.

## Materials and methods

### Computational analysis and selection of mutation from the TOPMed & ClinVar databases

At the time of this study, data from the Trans-Omics for Precision Medicine (TOPMed) program [28] were publicly available in its "freeze 5" version on the Bravo server [39]. This version

of the dataset consists of 463 million variants from 62,784 individuals and specifically contains 758 genomic variants and 124 predicted missense mutations in *KCNJ1*. To analyze the potential severity of the mutations, we ran a saturation mutagenesis analysis of ROMK with Rhapsody [30,31] available on a web interface (http://rhapsody.csb.pitt.edu/). We used a homology model of human ROMK (Uniprot #: P48048) obtained from Swiss-Model [148], which was built based on the crystal structure of Kir2.2 (PDB ID: 3SPG) [149]. Thus, Rhapsody was able to compute the pathogenicity probability, i.e., "score", only for amino acid residues 38–364 that are available in the homology model. A comprehensive list of all scores computed by Rhapsody for ROMK was deposited at https://github.com/mgm68/2023_ROMK_LoF/tree/main under the file name "Rhapsody_all_predictions_ROMK.". Because N377 is absent from the homology model, a Rhapsody score for the N377K mutation was instead obtained using a monomeric structure predicted by AlphaFold [80].

For further analysis, we prioritized mutations with a high Rhapsody score, i.e., more deleterious, as well as mutations located in regions previously found to be important for protein folding and channel function (see text for additional details). We also focused on mutations associated with Bartter syndrome that were classified clinically as being "of uncertain significance" in the ClinVar database [29]. Thus, T300R from ClinVar was added based on this classification, and also due to its position at residue T300 (since T300I had been selected from TOPMed).

## GWAS analysis on data from the UK Biobank

As noted in the **Results**, whole exome sequencing (WES) data from the UK Biobank [32] was used to perform three genome-wide association studies (GWAS). At the time of this analysis, the WES data had been released for ~200,000 individuals out of the ~500,000 total UKBB participants [61,62], among which there are 511 *KCNJ1* variants (**S4 Table**). After applying a minor allele frequency (maf) cutoff of >1e-5, the number of mutations used for the GWAS was 142. Phenotypic data were selected from the pool of the ~200,000 participants and included: (1) 25 disease phenotypes for relevance to ROMK function, Bartter syndrome type II, and hypertension [9,68] (e.g., systolic and diastolic blood pressure, serum urea, creatinine, calcium, and phosphate, and urine potassium and sodium), (2) 15 phenotypic codes, or "phecodes" [34], associated with Bartter syndrome type II, and (3) 168 continuous/quantitative metabolomics biomarkers. The quantitative phenotypes were normalized using inverse rank transformation to address non-normality of the underlying distribution [150].

The phecodes that were chosen represented 25 unique ICD10 codes relevant to Bartter syndrome, but individuals with phecodes related to diabetes and hypothyroidism were excluded from the analysis (see **Results**). Phecodes can be described as a mapping of grouping International Classification of Diseases (ICD) codes into clinically relevant groups [34,151]. Phecodes improve the power for association studies and enhance the accuracy of relevant phenotypes, in contrast to ICD codes. Specifically, we developed a custom algorithm to generate phecodes relevant to Bartter Syndrome based on an unsupervised multimodal automated phenotyping method [152]. The metabolomics biomarkers from the UK Biobank (data field category 220) were measured in plasma samples using a high-throughput NMR-based metabolic biomarker profiling platform developed by Nightingale Health Ltd.

The GWAS analyses were done using REVEAL: Biobank, a computational platform designed to explore, query, and perform large computations on biobank-scale datasets [35–38]. REVEAL: Biobank comprises R and Python application programming interfaces (API) for programmatic access to data and graphical user interfaces (GUI's) for selection of cohorts using phenotype and genotype filters, and then analyzes GWAS and Phenome Wide

Association Studies (PheWAS) results from a browser window. REVEAL: Biobank is built upon SciDB [51], a database solution ideal for storing and querying multi-omics data, utilizes elastic scaling through an application called BurstMode for efficient and cost-effective analyses and flexFS, a networked POSIX compliant filesystem for working with big data. REVEAL allows rapid and FAIR (a group of guiding principles for scientific data management [153]) access to the UK Biobank data, and multiple users can load, read, and write data in a secure, transactionally safe manner as data operations are guaranteed to be atomic and consistent (ACID compliant).

We used two algorithms, SAIGE (v0.44.6.5) [69] and REGENIE (v2.0.2) [70], to carry out the association analyses. Both algorithms are standards in GWAS bioinformatics workflows and are used to perform a regression test between a mutation of interest and a phenotype. Utilizing two algorithms also helped validate results. There were 12 covariates used in the GWAS: age, sex, and 10 genetic principal components provided by the UK Biobank (data field 22009).

The selection of alleles for further characterization is described in the **Results**.

### Plasmid construction

Rat ROMK1 was amplified from the pSPORT1-ROMK1 vector [154] and inserted into the yeast expression vector pRS415 with SmaI and XhoI and was flanked by the *TEF1* promoter and *CYC1* terminator [155], as described [15,31]. Point mutations in *KCNJ1* were introduced into the resulting pRS415TEF1-ROMK1 vector using either two-step overlap extension mutagenesis [156] or site-directed mutagenesis with the QuikChange kit (Agilent Technologies, CA, USA, catalog # 200523). To express ROMK variants in HEK293 cells, the DNA inserts were digested with BamHI and XhoI from the yeast vector and subcloned into pcDNA3.1(+). The DNA sequences of all variants in the ROMK inserts were confirmed by Sanger sequencing (GENEWIZ, S Plainfield, NJ, USA). All primers used in this study are listed in **S8 Table**.

### Yeast strains and growth conditions

A *Saccharomyces cerevisiae* strain lacking the Trk1 and Trk2 potassium transporters, *trk1Δtrk2Δ*, was employed to assess mutation severity by measuring the ability of each mutation to restore growth on low potassium medium, as described previously [15,31,51,53]. Briefly, plasmids were transformed into yeast via the standard lithium-acetate method [157], and yeast were grown at 30°C in liquid or solid synthetic complete (SC) medium lacking leucine, which contained monosodium glutamate as the main nitrogen source and buffered to pH 4.5 with MES. Media was supplemented with either 100 mM or 25 mM KCl. Due to the presence of residual potassium in the agar and nitrogen source, each plate contained an additional 7–10 mM KCl [49,158].

To perform protein stability assays in yeast (see below), we utilized the indicated yeast strains (i.e., *trk1Δtrk2Δ* and *pdr5Δ*; see **S8 Table**). Cells were grown at 30°C and switched to 37°C at the beginning of the chase. Assays using *CDC48* and the isogenic *cdc48-2* strains were propagated at 26°C and then shifted to 39°C (**S8 Table**).

### Yeast viability assays

Yeast viability assays were conducted as described [31]. For serial-dilution growth assays on solid medium, saturated overnight cultures were diluted to an $A_{600}$ of 0.20, then further diluted 5-fold four times in a standard 96-well plate, followed by inoculation into SC-Leu medium supplemented with 100 mM or 25 mM KCl using a 48-pin replica plater (Sigma-Aldrich, St. Louis, MO, USA). Plates were incubated at 30°C and imaged after two days with the Bio-Rad ChemiDoc XRS+ imager. For assays in liquid medium, saturated overnight cultures were

diluted to an $A_{600}$ of 0.20 with SC-Leu medium containing 25 mM KCl in a 96-well plate. The plates were then covered with a Breathe-Easy gas permeable membrane (Diversified Biotech, Dedham, MA, USA), and cell density readings were recorded using the Cytation 5 plate reader (BioTek, Winooski, VT, USA) every 30 min for the indicated time with constant shaking at 30˚C.

## Yeast stability assays

Stability assays in yeast were carried out based on established protocols [15,53], with minor modifications. In brief, yeast cultures transformed with the ROMK expression vector (see above) were grown in selective media to mid-log phase ($A_{600} = 0.7–1.5$), diluted to the same density (typically $A_{600} = 1.0$), and transferred to a water bath with constant shaking at 200 rpm. The cells were then incubated for 30 min at 30˚C (*trk1Δtrk2Δ*) or for 2 hr at 39˚C (the *CDC48* and *cdc48-2* strains). A similar protocol was followed for when the *pdr5Δ* strain was employed, except the initial 30 min incubation was performed in the presence of 50 μM MG-132 or an equal volume of the vehicle (DMSO). Next, cycloheximide was added to a final concentration of 150 μg/ml, at which point a 1 ml aliquot was collected. Subsequent 1 ml aliquots were collected at the indicated time points, flash frozen in liquid $N_2$, and either kept at -20˚C or were subject to immediate processing and lysis.

The levels of ROMK at each time point were assayed as previously outlined [15,51]. After lysis in 300 mM NaOH, 1% β-mercaptoethanol, 1 mM PMSF, 1 μg/ml leupeptin, and 0.5 μg/ml pepstatin A, total protein was precipitated with 5% trichloroacetic acid on ice. The mixture was the centrifuged at 14,000 rpm for 10 min at 4˚C in a microfuge and subject to SDS-PAGE and immunoblot analysis. See **S8 Table** for more information on the antibodies and dilutions used.

## HEK293 cell culture, transfection, and stability assays

HEK293 cells (Thermo Fisher, Waltham, MA, USA) were cultured at 37˚C in Dulbecco's Modified Eagle's Medium containing high levels of glucose (Sigma-Aldrich, St. Louis, MO, USA) and supplemented with 10% Fetal Bovine Serum and a mixture of penicillin/streptomycin (final concentration: 500 units/ml). Cells in 6-well dishes (passage 2–3, 60–90% confluency) were transfected with 2 μg of plasmids carrying the indicated ROMK mutants using Lipofectamine 2000 (Invitrogen, Waltham, MA, USA), and the media was replaced after 4 hr. Protein stability was measured based on an established protocol, with slight modifications [15]. In short, fresh media containing 50 μM MG-132 or the equivalent volume of DMSO was added 18–20 hr post transfection. After a 30 min incubation, a final concentration of 50 μg/ml cycloheximide was introduced, and cells were collected at the indicated time points. For steady state measurements after treatment with CB-5083, a slightly modified protocol was followed. In brief, HEK293 cells were cultured in 12-well dishes and transfected with 0.6 μg of the indicated ROMK expression vector, and 20 hr post transfection, the media was replaced with media containing 50 μg/ml cycloheximide in the presence or absence of a final concentration of 20 μM CB-5083. Cells were collected after a 4 hr incubation at 37˚C. In both assays, cell pellets were collected and retained at -20˚C.

Cells were lysed in TNT buffer (50 mM Tris, pH 7.4, 150 mM NaCl, 1% Triton X-100) supplemented with a protease inhibitor cocktail (Roche, Basel, Switzerland) on ice for 20 min with occasional agitation. The mixture was then centrifuged at 13,000 rpm for 10 min at 4˚C in a microfuge to remove the nuclear fraction, and the supernatant was transferred into new tubes and SDS sample buffer containing 150 mM DTT was then added to facilitate protein analysis by SDS-PAGE and immunoblots, as described [15,51].

## Cell-surface biotinylation assays

Cell-surface biotinylation assays were performed as published [51], with minor modifications. In short, 20–22 hr post transfection, HEK293 cells expressing the indicated ROMK construct were treated with a final concentration of 125 µg/ml cycloheximide for 2 hr at 37˚C. The plates were then transferred onto ice, washed three times, and treated with 0.3 mg/ml EZ-Link Sulfo-NHS Biotin (Thermo Fisher, Waltham, MA, USA) for 1 hr. Excess biotin was quenched by washing the cells with 100 mM glycine two times, and then the cells were lysed in 20 mM HEPES, pH 7.6, 1 mM EDTA, 1 mM EGTA, 25 mM NaCl, 1% Triton-X, 10% glycerol containing a protease inhibitor cocktail (Roche, Basel, Switzerland) for 1 hr before the mixture was centrifuged cold at 14000 rpm for 15 min to remove any insoluble material. The concentration of the liberated soluble protein was assessed with the Pierce BCA protein assay kit (Thermo Fisher, Waltham, MA, USA), and equal amounts of protein (180–250 µg) were brought to a total volume of 1 ml in the same buffer as above. After an aliquot corresponding to 1% of the total was collected, the remaining protein was added to 30 µl of Pierce NeutrAvidin-agarose beads (Thermo Fisher, Waltham, MA, USA) and incubated overnight at 4˚C. The next day, the beads were washed three times and subject to SDS-PAGE and immunoblot analysis (see **S8 Table**).

## Two-electrode voltage clamp measurements

pRS415-ROMK expression plasmids (see above) were linearized and used as templates for cRNA synthesis by *in vitro* transcription using T3 RNA Polymerase (Ambion, Inc., Life Technologies, Carlsbad, CA, USA). The resulting cRNAs were then purified with an RNA purification kit (Qiagen, Hilden, Germany), quantified, and the cRNA quality was assessed by denaturing agarose gel analysis.

Oocytes from *Xenopus laevis* were harvested with a protocol approved by the University of Pittsburgh's Institutional Animal Care and Use Committee. Briefly, stage V and VI oocytes were treated with collagenase type II and trypsin inhibitor to remove the follicle cell layer. Oocytes were then injected with 1 ng of the indicated cRNA and incubated at 18˚C in a slightly modified Barth's solution (15 mM HEPES, pH 7.4, 88 mM NaCl, 10 mM KCl, 2.4 mM NaHCO$_3$, 0.3 mM Ca(NO3)$_2$, 0.41 mM CaCl$_2$, 0.82 mM MgSO$_4$, 10 µg/ml streptomycin sulfate, 100 µg/ml gentamycin sulfate) for 20–30 hr. Next, two electrode voltage clamp experiments were performed at room temperature (20–24˚C) with the TEV200A Voltage Clamp Amplifier (Dagan Corporation, Minneapolis, MN, USA) and the DigiData 1440A and Clampex 10.4 software (Molecular Devices, San Jose, CA, USA). Oocytes were placed in a recording chamber and perfused with a bath solution (10 mM HEPES, pH 7.8, 50 mM KCl, 48 mM NaCl, 2 mM CaCl$_2$, 1mM MgCl$_2$) at a constant flow rate of 5–10 ml/min. Whole-cell currents were recorded at a series of voltages (-160 and 100 mV in 20 mV increments), in the absence and presence of 1mM BaCl$_2$ in the bath solution. Data were analyzed using Clampfit in the pClamp 10.4 package. Ba$^{2+}$-sensitive currents, which represent ROMK channel activity in oocytes [159], were defined as the difference in currents measured in the absence and presence of BaCl$_2$. Ba-sensitive currents were quantified, and graphs were made using GraphPad Prism (ver. 9.5.0).

## Receiver operating characteristic (ROC) curve

To assess how well Rhapsody distinguishes between deleterious and neutral mutations, we calculated a receiver operating characteristic (ROC) curve for the 17 mutations we selected from TOPMed and ClinVar (**Fig 1**), along with the previously published Y314C mutation [15]. We first ordered all candidate mutations from best to worst by their Rhapsody scores. We then

considered all possible partitions (cutoffs) of this ordered list. For each cutoff, mutations ranking above the cutoff were tentatively classified as deleterious, and those below were classified as neutral. We compared these provisional classifications to the corresponding experimental assessments to calculate false positive and true positive rates for each cutoff and constructed a ROC curve by plotting each false-positive-rate/true-positive-rate data point. Specifically, we assigned a "true positive" to the ROMK mutants that had been experimentally verified as severe or moderate (see **S3 Table**), i.e., mutations that resulted in a relative endpoint OD of $< 0.9$. In contrast, mutants above the cutoff with endpoint ODs of $\geq 0.9$ are designated "false-positive". After computing the ROC curve, we then calculated the area under the ROC curve (AUROC or AUC) using the composite trapezoidal rule. For the ROC curve of Rhapsody with incorporated data from yeast growth assays using the ROMK-K80M construct, the designations for two mutations (F93V and V122E) were changed to deleterious based on the growth defects (**S1 Fig**). Similar analyses were performed to compute ROC curves and AUROC values for Polyphen-2, EVmutation, and EVE. For EVmutation, no score was obtained for the M357T mutant, and thus this allele was omitted from the analysis. A detailed table of the pathogenicity scores and true/ false-positive assessment for each method is under the name "Simplified ROC analysis" in this depository: https://github.com/mgm68/2023_ROMK_LoF/tree/main.

## Statistical methods

For the stability assays, CB-5083 treatments and cell-surface biotinylation in HEK293 cells, p-values were calculated with a two-tailed Student's t-test for independent samples. In the two-electrode voltage clamp experiments, statistical analysis was conducted using Kruskal-Wallis and Dunn's multiple comparisons tests, and normality was examined with Shapiro-Wilk tests.

## Supporting information

**S1 Fig. Growth assays of yeast expressing select TOPMed mutations in the context of the K80M allele.** The growth of yeast containing a vector control, wild-type ROMK, or the indicated mutation in the context of an activating mutation, K80M, was measured in liquid medium containing 10mM KCl. Note that the growth phenotype of a mutant should be compared to the "ROMK-K80M" curve at the top. $OD_{600}$ readings were recorded every 30 min for 23.5 hrs. Data represent the means of 8 replicates, ± S.E (error bars). A summary of these data is shown in **S3 Table**. The top graph contains the growth curves for all variants, but since many overlap at OD ~0.1 (red scale bar), this section of the graph was magnified for clarity (see graph at the bottom).
(DOCX)

**S2 Fig. REVEAL: Biobank platform.** REVEAL: Biobank is a platform built upon SciDB, a computational database ideal for large scale linear algebra operations, and is comprised of an R-programmed API and Graphic User Interfaces (GUIs) for cohort selection and PheWas visualization. This platform has multiple features: elastic scaling (Burst Mode) for efficient and cost-effective analyses; Bridge, a cloud-optimized array format; and flexFS, a networked POSIX compliant filesystem for working with big data in the UK Biobank. See **Materials and Methods** for details.
(DOCX)

**S3 Fig. The degradation of select ROMK mutants is Cdc48-dependent in yeast.** Stability assays performed in yeast expressing the wildtype ROMK, or ROMK carrying the mutation G228E, L320P, or N377K. To assess the effect of the yeast AAA$^+$-ATPase Cdc48 on protein

degradation, a temperature sensitive yeast strain (*cdc48-2*) was used. Yeast cultures were grown to mid-log phase (OD$_{600}$ 0.7–1.5) at a permissive temperature, diluted, and incubated at a non-permissive temperature of 39˚C for 2 hours before adding cycloheximide. Cells were then processed, and immunoblot analysis was performed (see **Materials and Methods**). Representative immunoblots are shown, and graphs show the percentage of the protein remaining over time, compared to the 0 min (m) time point, as quantified by ImageJ (ver. 1.53c). Graphs were made using GraphPad Prism (ver. 9.5.0), and data represent the means of at least three independent experiments, ± S.E. (error bars). For each experiment, a representative immunoblot is shown.
(DOCX)

**S4 Fig. Select mutants reduce ROMK protein levels at the cell surface.** Representative cell-surface biotinylation assay showing the surface expression levels of the indicated ROMK variants expressed in HEK293 cells. The experimental set-up is identical to the experiment shown in **Fig 7**, except data for N377K are included.
(DOCX)

**S5 Fig. Structural modeling suggests the T300R mutation occludes the cytoplasmic pore.** Homology model shows the pore of the tetrameric ROMK channel, as viewed from the cytoplasmic side. Potassium ions (spheres in the center of the pore) are shown in salmon and outlined with dotted black lines. Spheres depicting the positions of T300 **(A)**, T300I **(B)**, and T300R **(C)** are in green, cyan, and magenta, respectively. Only residues 184–364 of each chain are shown for clarity. The homology model was built based on the crystal structure of Kir2.2 (PDB ID: 3SPG), which is 47.42% identical to ROMK1. Images were rendered using PyMOL (ver. 2.6.0).
(DOCX)

**S6 Fig. ROC curves of TOPMed and ClinVar mutations in relation to their pathogenicity scores and yeast growth phenotypes.** ROC curves (in blue solid lines) were computed for the 17 selected mutations from TOPMed and ClinVar, along with a previously published Y314C mutation [15]. True positive and false positive designations were determined based on yeast growth data from **Fig 2** and **S3 Table**. Mutations were considered deleterious if they resulted in a relative endpoint of OD600 <0.9. Otherwise they were considered neutral. The area under the ROC curve (AUROC or AUC) were then calculated using the composite trapezoidal rule. Five ROC curves with their corresponding AUROC were computed: **(A)** Rhapsody using yeast growth data from **Fig 2**; **(B)** Rhapsody, with both the original yeast growth dataset and the new data using the ROMK-K80M construct incorporated. In particular, the newly observed growth defects changed the designations of two mutations (F93V and V122E) to deleterious; **(C)** Polyphen-2; **(D)** EVmutation; **(E)** EVE. For reference, a line of no-discrimination (dotted black line) is shown, which corresponds to a purely random classifier.
(DOCX)

**S7 Fig. Slow growth *trk1Δtrk2Δ* yeast colonies fail to propagate on a nonfermentable carbon source.** Large and small colonies of *trk1Δtrk2Δ* yeast carrying an empty vector, the ROMK protein, or the N377K mutant were propagated on synthetic complete medium lacking leucine and containing 3% glycerol. Plates show two independent replicates, and images were taken with the Bio-Rad ChemiDoc XRS+ imager after a 7-day incubation (4 days at 30˚C, 3 days at 22˚C).
(DOCX)

**S8 Fig. Heterozygous ROMK mutants exhibit an intermediate phenotype in whole-cell currents in *X. laevis* oocytes.** Currents recorded by two-electrode voltage clamps (TEVC) in *X. laevis* oocytes as described in **Fig 8** and **Materials and Methods**. To recapitulate

heterozygosity, oocytes were co-injected with 0.5 ng each of wild-type (WT) ROMK and the indicated mutant. (A) Graph shows the $Ba^{2+}$-sensitive/ROMK current in oocytes injected with the indicated cRNAs, as recorded by TEVC. (B) Normalized currents, which are defined as $Ba^{2+}$-sensitive currents divided by the means of the wild-type currents. Error bars in the graphs show the means of all replicates (in parentheses), ±S.D. p-values (shown above the data) were computed using Kruskal-Wallis and Dunn's multiple comparisons tests.
(DOCX)

**S1 Table. Comprehensive list of ROMK missense mutations in the TOPMed database.** Table shows the Rhapsody scores and predictions of 124 ROMK missense mutations from the TOPMed program that were analyzed in this study. The analysis was performed using a ROMK homology model that contains amino acids 38–364 (see **Fig 1**), so any residue outside of this range lacks a Rhapsody score and is designated "-". "Del" indicates a substitution is predicted to be deleterious, whereas "Neu" (neutral) is predicted to have no effects on channel function. A designation of "Prob. Del" or "Prob. Neu" indicates that the Rhapsody score is close to the 0.5 cutoff for being deleterious. For example, the Rhapsody scores of I85N and F94S are 0.512 and 0.470, and thus, these mutations are categorized as "Prob. Del" and "Prob. Neu", respectively. The mutations listed in **bold** were selected for growth analysis in yeast.
(DOCX)

**S2 Table. Targeted list of 17 mutations showing Rhapsody scores, predicted phenotypes, and background information.** Rhapsody was used to predict ROMK mutation severity based on structural, evolutionary, and dynamic features. The analysis was performed with a tetrameric ROMK homology model (Uniprot number: P48048), which was built in Swiss-Model [148] based on the crystal structure of Kir2.2 (PDB ID: 3SPG). A Rhapsody pathogenicity probability (or "Rhapsody score") was computed for each mutation, and a "Del" (deleterious) denotation was assigned if the probability is ≥ 0.5, whereas a "Neu" (neutral) indicates a probability of < 0.5. "Prob. Del" denotes that the Rhapsody probability is close to the 0.5 deleterious cutoff (i.e., P185S probability score is 0.549). * denotes an uncharacterized Bartter mutation, which was defined as a disease-associated mutation in ClinVar, but is listed as having uncertain clinical significance. ¶ denotes the mutation obtained from ClinVar.
(DOCX)

**S3 Table. Growth phenotype summary of yeast expressing TOPMed and ClinVar mutations.** The Rhapsody score (i.e., probability) and prediction for each of the 17 mutations, along with their growth phenotypes in medium supplemented with low potassium. For columns 4 and 5 ("ROMK"), the growth assays of yeast expressing the mutations in the context of wild-type (WT) ROMK were conducted in 25mM KCl, and the "end-point" $OD_{600}$ recordings at 48 hrs were normalized to WT. Each growth phenotype assessment shown in column 5 was based on the results of up to three independent experiments (representative graphs shown in **Fig 2**). The categorization of the growth defects was determined based on the normalized endpoint $OD_{600}$ (to WT ROMK), and are defined as follows: No defect, OD ≥ 1 (1 = 100% WT); Slight defect, 0.9 ≤ OD < 1 (90–100% WT); Moderate defect: 0.8 ≤ OD < 0.9 (80–90% WT); Severe defect: OD < 0.8 (80% WT). The data in columns 6 and 7 ("ROMK-K80M") represent growth phenotypes of yeast co-expressing the indicated allele and an activating mutation, K80M, in 10mM KCl medium (also see **S1 Fig**). The end-point $OD_{600}$ values at 23.5 hrs and slopes of each of the growth curves ("Max V", as calculated by the Gen5 software, BioTek Instruments, ver. 3.12) were normalized to the ROMK-K80M control. Data represent the means of 8 replicates. * denotes an uncharacterized Bartter mutation, and ¶ denotes a mutation in ClinVar. Additionally, a "-"marks where data were absent.
(DOCX)

**S4 Table. ROMK variants from the UK Biobank analyzed in this study.** Table shows 511 *KCNJ1* variants available in the whole-exome sequencing (WES) database, which contains data from ~200k participants from the UK Biobank [61,62]. Columns represent the chromosomal location and the nucleotide change for each substitution, as well as their minor and alternative allele frequencies (denoted as maf and aaf, respectively).
(XLSX)

**S5 Table. Phenotypes examined in association analyses of the 511 *KCNJ1* variants in the UK Biobank.** Phenotypes were selected from the pool of the ~200,000 participants in the UK Biobank and included three groups: (1) 25 disease phenotypes for relevance to ROMK function, Bartter syndrome type II, and hypertension [9,68] (e.g., systolic and diastolic blood pressure, serum urea, creatinine, calcium, and phosphate, and urine potassium and sodium), (2) 15 phenotypic codes, or "phecodes" [34], associated with Bartter syndrome type II, and (3) 168 continuous/quantitative metabolomics biomarkers. Each group of phenotypes was listed in one tab of the Excel file.
(XLSX)

**S6 Table. Genotype distribution in the UK Biobank of ROMK variants with significant associations with disease phenotypes.** The top 5 rows of the table show the population distribution of binary disease phenotypes, i.e., "phecodes", and the bottom 9 rows show the distribution of the metabolite disease phenotypes. "Hom." denotes the number of individuals homozygous for the indicated mutation, "Het." means heterozygous, and "WT" stands for wildtype, i.e., individuals without the indicated mutation. For the binary phenotypes, both the number of controls and cases are listed.
(DOCX)

**S7 Table. Pathogenicity predictions of 17 TOPMed and ClinVar mutations made by different computational methods.** Table shows the pathogenicity predictions made by the indicated computational tools for the 17 TOPMed and ClinVar mutations. The second column shows the phenotype exhibited by each mutant when expressed in yeast, as shown in **S3 Table**. The growth phenotype noted with an asterisk (*) indicates that a growth defect was only observed when the mutant was expressed in the context of the K80M allele (**S1 Fig**). In the remaining columns, a mutation that is predicted to be pathogenic is marked with "Del" (deleterious), while a benign mutation is designated "Neu" (neutral). Whether there was an uncertainty in the prediction, or a prediction is unavailable, is also indicated ("Uncertain" or "N/A"). In the bottom row, an accuracy assessment for each method based on yeast growth phenotype is provided. A "slight" growth defect was counted for both deleterious and neutral. The computational tools employed are Rhapsody [40], Polyphen-2 [41], Evmutation [42], EVE [125], SNPs&GO [161].
(DOCX)

**S8 Table. Primers, yeast strains, and antibodies used in this study.**
(DOCX)

## Acknowledgments

We thank Luca Ponzoni and Ivet Bahar for valuable technical and scientific discussion, Paul Welling for generously providing the anti-ROMK antiserum, David Everman and Anne Childers for scientific discussion, and members of the Brodsky and O'Donnell labs.

## Author Contributions

**Conceptualization:** Nga H. Nguyen, Srikant Sarangi, Thomas R. Kleyman, Zachary W. Pitluk, Jeffrey L. Brodsky.

**Data curation:** Nga H. Nguyen, Srikant Sarangi, Erin M. McChesney, Shaohu Sheng, Jacob D. Durrant.

**Formal analysis:** Nga H. Nguyen, Srikant Sarangi, Shaohu Sheng, Jacob D. Durrant, Aidan W. Porter, Thomas R. Kleyman.

**Funding acquisition:** Thomas R. Kleyman, Zachary W. Pitluk, Jeffrey L. Brodsky.

**Investigation:** Nga H. Nguyen, Srikant Sarangi, Erin M. McChesney, Shaohu Sheng, Jacob D. Durrant, Aidan W. Porter.

**Methodology:** Nga H. Nguyen, Srikant Sarangi, Shaohu Sheng.

**Project administration:** Zachary W. Pitluk, Jeffrey L. Brodsky.

**Software:** Srikant Sarangi, Zachary W. Pitluk.

**Supervision:** Thomas R. Kleyman, Zachary W. Pitluk, Jeffrey L. Brodsky.

**Visualization:** Nga H. Nguyen.

**Writing – original draft:** Nga H. Nguyen, Srikant Sarangi, Shaohu Sheng, Jeffrey L. Brodsky.

**Writing – review & editing:** Aidan W. Porter, Thomas R. Kleyman, Zachary W. Pitluk, Jeffrey L. Brodsky.

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
