## [Decision Letter · Decision Letter 0]

22 Sep 2023

Dear Dr Brodsky,

Thank you very much for submitting your Research Article entitled 'Genome mining yields putative disease-associated ROMK variants with distinct defects' to PLOS Genetics.

The manuscript was fully evaluated at the editorial level and by independent peer reviewers. The reviewers appreciated the attention to an important topic but identified some concerns that we ask you address in a revised manuscript.

We therefore ask you to modify the manuscript according to the review recommendations. Your revisions should address the specific points made by each reviewer.

Yours sincerely,

Anne O'Donnell-Luria, MD, PhD

Academic Editor

PLOS Genetics

Hua Tang

Section Editor

PLOS Genetics

Thank you for taking the time to further revise your manuscript to meet the previous rounds of reviews. It is substantially improved though there are a few suggestions from the reviewers that would be good to incorporate.

Reviewer's Responses to Questions

**Comments to the Authors:**

Reviewer #1: In their revised manuscript, the authors have made several additions and revisions that have significantly strengthened the study. For example, the addition of new experiments analyzing mutations in the context of the hyperactive K80M allele to enhance the dynamic range of the growth phenotype serves to strengthen the manuscript. Overall, this is an excellent study that is well-written and will appeal to the readership of PLOS Genetics. I have only a couple of very minor comments on the revised manuscript:

Minor point:

1. Inclusion of the end point OD600 in Table S3 is very helpful. However, it is still unclear to me what thresholds are being used for each growth defect category, particularly in the case of the “slight defect” mutants. For example, A214V has an end point OD of 0.99, compared to 1.00 for the ROMK positive control. Does this really constitute a “slight difference”? I still can’t find a clear description of how mutants were categorized in the main text or in the legend of either Figure 2 or Table S3. It looks like it was all based on end point OD, with a moderate defect corresponding to 80-90% of WT, while a slight defect is 90-99% of WT. Just clarifying this in the text or one of the legends would be very helpful.

Minor edits:

1. Lines 76-79: this sentence is repetitive and needs to be revised.

2. Line 442: I think “datamining” should be two words

Reviewer #2: In "Genome mining yields putative disease-associated ROMK variants with distinct defects", Nguyen et al. present a revised version of a previous manuscript on the same topic. The authors have extensively commented on the reviewer comments and clearly put in a lot of additional work to improve the manuscript. Below are some comments that would be good to clarify for a final version of the article.

- The abstract mentions "a computational algorithm that predicts protein misfolding and disease severity" - it would be good to name the algorithm, probably Rhapsody? Also, is it really misfolding (following up on the first round of reviews, where this question also came up) that is assessed? The term keeps popping up throughout the manuscript although the response to reviewers mentions that indeed Rhapsody is not folding-specific, so it's unclear if additional edits are missing here.

- Overall, it should be clear throughout the manuscript when protein misfolding is meant (e.g. from prior work focusing on that) and when general pathogenicity prediction or assessment of function is carried out.

- It should further be clarified if Rhapsody assesses disease severity - via a quantitative score - or is mainly a binary assessment of whether a variant is pathogenic or not, which is how it appears to be used in Table S7.

- Does Rhapsody work on AlphaFold models? Are the results for ROMK the same on the AF2 model vs. the homology model?

- Figure S6, ROC curve. It is a little bold to call an AUC of 0.76 "high predictive power". Another challenge is that there are very few datapoints, as seen from the stair-like or rugged nature of the curve. The results section mentions 124 datapoints for ROMK, are these all included here? A ROC curve based on all 127 datapoints is much more convincing than one based on the much smaller subset of variants experimentally tested in this work. Further, the authors should include the ROC curves for other predictors like PolyPhen2 and EVmutation (see also Table S7) to visualise the differences in performance.

In the context of the comments raised by reviewer 3 wrt heterozygous phenotypes, https://doi.org/10.1016/j.gendis.2015.02.008 may offer some insights.

Reviewer #3: My concerns have been adressed.

**Have all data underlying the figures and results presented in the manuscript been provided?**

Reviewer #1: Yes

Reviewer #2: None

Reviewer #3: **No: **

PLOS authors have the option to publish the peer review history of their article (what does this mean?). If published, this will include your full peer review and any attached files.

Reviewer #1: **Yes: **Jason MacGurn

Reviewer #2: No

Reviewer #3: No

---

## [Editor Report · Decision Letter 1]

4 Nov 2023

Dear Dr Brodsky,

We are pleased to inform you that your manuscript entitled "Genome mining yields putative disease-associated ROMK variants with distinct defects" has been editorially accepted for publication in PLOS Genetics. Congratulations! Thank you for your persistence in continuing to respond to reviewers comments to improve the quality of the manuscript. 

Yours sincerely,

Anne O'Donnell-Luria, MD, PhD

Academic Editor

PLOS Genetics

Hua Tang

Section Editor

PLOS Genetics

Comments from the reviewers (if applicable):

**Data Deposition**

http://datadryad.org/submit?journalID=pgenetics&manu=PGENETICS-D-23-00911R1

**Press Queries**

---

## [Editor Report · Acceptance letter]

9 Nov 2023

PGENETICS-D-23-00911R1 

Genome mining yields putative disease-associated ROMK variants with distinct defects 

Dear Dr Brodsky, 

We are pleased to inform you that your manuscript entitled "Genome mining yields putative disease-associated ROMK variants with distinct defects" has been formally accepted for publication in PLOS Genetics! Your manuscript is now with our production department and you will be notified of the publication date in due course.

With kind regards,

Anita Estes

PLOS Genetics

On behalf of:
